# Decisions bias future choices by modifying hippocampal associative memories

Lennart Luettgau [1,2 ✉], Claus Tempelmann[3], Luca Franziska Kaiser [1,2] & Gerhard Jocham[1,2]

Decision-making is guided by memories of option values. However, retrieving items from memory renders them malleable. Here, we show that merely retrieving values from memory and making a choice between options is sufficient both to induce changes to stimulus-reward associations in the hippocampus and to bias future decision-making. After allowing participants to make repeated choices between reward-conditioned stimuli, in the absence of any outcome, we observe that participants prefer stimuli they have previously chosen, and neglect previously unchosen stimuli, over otherwise identical-valued options. Using functional brain imaging, we show that decisions induce changes to hippocampal representations of stimulus-outcome associations. These changes are correlated with future decision biases. Our results indicate that choice-induced preference changes are partially driven by choice-induced modification of memory representations and suggest that merely making a choice - even without experiencing any outcomes - induces associative plasticity.

[1] Biological Psychology of Decision Making, Institute of Experimental Psychology, Heinrich Heine University Düsseldorf, Düsseldorf, Germany. [2] Center for Behavioral Brain Sciences, Otto-von-Guericke University, Magdeburg, Germany. [3] Department of Neurology, Otto-von-Guericke University, Magdeburg, Germany. ✉email: luettgau@hhu.de

According to neo-classical economic models of decision-making, choices are guided by memories of option values[1]. This unidirectional view has been challenged by cognitive accounts of decision-making, suggesting that memory representations of option values might themselves be subject to changes induced by an agent's choices. This suggests a bidirectional relationship between value representations in memory and decision-making[1,2].

Even though real-life decisions often involve memory retrieval of learned associations between reward-predictive cues and outcomes, memory mechanisms underlying choice-induced preference changes have, to the best of our knowledge, never been systematically studied[1,3–5]. This might be partially related to the fact that most studies on choice-induced preference changes employed direct presentation of the outcomes to be chosen. However, this approach by design obliterates and confounds underlying associative learning contributions to the revaluation process[6], and is blind to related memory processes, such as retrieval competition[7].

In naturalistic decision-making scenarios, choices often have to be made without direct experience of feedback. Instead, decision makers have to rely on relational knowledge of actions and outcomes. Likely candidate mechanisms for behavioral adaptation without direct external feedback are memory retrieval dynamics. It is well established that retrieval of an item from memory, e.g. a conditioned stimulus (CS) triggering retrieval of an associated outcome, leads to improved remembering. However, memory for competing items, e.g. another CS associated with the same outcome, is impaired simultaneously[7–9]. Such retrieval-induced forgetting[8] would predict choice biases towards previously chosen CS based on retrieval-related strengthening of CS–US association. However, the same effect would be predicted for a previously presented, but unchosen CS, since both chosen and unchosen CS activate neural populations representing the associated outcome[10–15]. A recent theoretical framework[16] suggests nonmonotonic plasticity during associative memory retrieval: Inactive memories remain unaltered, moderately activated associative memories are weakened, and high activation strengthens memories[16]. Translating this idea to memory-based decisions between two CS, we assume that both CS will moderately activate neural populations representing the associated outcome (as the outcome is never presented). However, consistent with the finding that chosen options receive higher attentional weighting than unchosen options (as reflected in learning rates[17,18]), we further assume that choices of a CS will induce additional activation of the associated outcome, whereas this will not be the case for unchosen CS, retaining an intermediate activation state of the associated outcome. Thus, we hypothesize that choosing a CS will strengthen the related stimulus-outcome association. Conversely, not choosing a CS will weaken the respective stimulus-outcome association. We expect that these choice-induced alterations of the associative memory structure will result in subsequent preference changes.

We expect choice-induced preference changes to be driven by modifications of stimulus-outcome associations in the hippocampus and lateral orbitofrontal cortex, two key regions for storing and updating associative representations[10,14].

Thus, the goal of the present study is twofold. First, we aim at investigating how choice-related alterations of associative memories bias future decision-making. Second, we seek to investigate a neurobiologically plausible mechanism underlying choice-induced preference changes. To test our key predictions, we designed a learning and decision-making paradigm which we use in three independent behavioral experiments and one functional magnetic resonance imaging (fMRI) experiment. For the fMRI study, we exploit repetition suppression (RS) effects[10,11,19–21] to measure associative strength between conditioned (CS) and unconditioned stimuli (US)[10,14]. Participants first establish Pavlovian associations between CS and differently valued US. Next, in a choice-induced revaluation, participants make binary choices between differently valued CS, without observing the associated US. Finally, in a probe phase, where they make choices between all possible CS combinations, participants show preference increases for previously chosen, and preferences decreases for previously unchosen CS, compared to otherwise equivalent CS. These choice-induced alterations in decision behavior are accompanied by corresponding changes in CS–US RS effects in the hippocampus and lateral orbitofrontal cortex. Our findings are corroborated by multivariate pattern similarity analyses (a variant of representational similarity analysis, RSA[22]). Furthermore, the magnitude of the hippocampal RS effect correlates with individual probe phase decision biases.

## Results

**Behavioral experiments**. First, we detailed the behavioral choice-induced revaluation effect in three independent experiments. In each experiment, participants learnt associations between neutrally rated CS[23] and three food items[24] serving as unconditioned stimuli (US, Fig. 1a). For each participant, the US were individually chosen based on a prior rating of subjective preference, and a low-value (US−), an intermediate-value (US0), and a high-value (US+) food item was selected. Next, participants rated kanji stimuli[23] according to their subjective preference. Six of these kanjis rated in close proximity to neutral were selected as CS for Pavlovian learning. Two CS each were paired with one US, resulting in three categories of differently valued CS: $CS_{A/B}^+$, $CS_{A/B}^0$, and $CS_{A/B}^-$, for high-value, intermediate-value, and low-value CS.

The Pavlovian learning phase was followed by choice-induced revaluation. From two value categories, one CS each was selected, and participants made binary choices between them, interspersed with lure decisions between non-reward-predictive kanjis (Fig. 1b). Crucially, no associated US were presented following choices, excluding the possibility of alterations in strength of stimulus-outcome associations due to directly experienced outcomes. The choice-induced revaluation phase was followed by a decision probe phase in which participants chose repeatedly between all binary CS combinations to assess preferences (Fig. 1c). Again, no outcomes were presented. The key comparison was between CS presented during revaluation and CS from the same value category that had not been presented.

**Decisions are biased by past choices**. There was evidence for value transfer from US to CS across all studies (Fig. 1e–h), as indicated by significant main effects of CS value on probe phase choice probabilities (CP, all $Fs > 94.99$, $Ps < 0.001$, $\eta_p^2 s > 0.69$, $1 - \beta s > 0.99$, repeated-measure analyses of variance, rmANOVA). Decision-making during the revaluation phase had clearly dissociable effects on choices during the probe phase. CS that were chosen during the revaluation phase were more likely to be selected during the later probe phase compared to the CS of equal value that were not presented during choice-induced revaluation.

In Experiment 1, participants ($N = 40$) made choices between the intermediate-value $CS_A^0$ and the high-value $CS_A^+$ during the choice-induced revaluation phase. As we had directed hypotheses for the choice effects (increased CP for the previously chosen and decreased CP for the previously unchosen CS), we used one-tailed post hoc tests. In the probe phase, participants preferred $CS_A^+$, the previously selected stimulus, compared to $CS_B^+$ ($Z = 3.98$,

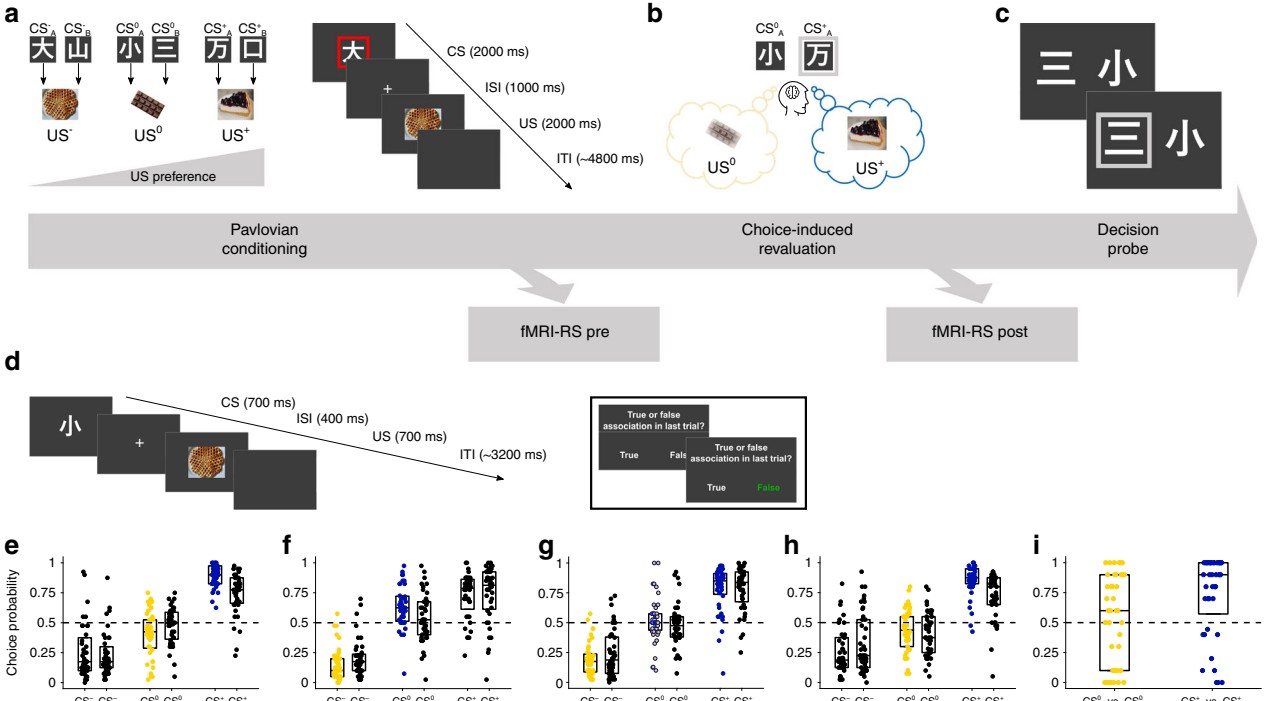

**Fig. 1 Task schematic and behavioral results. a** Participants rated subjective desirability of conditioned stimuli (CS, kanjis) and unconditioned stimuli (US, food items). Kanji images were selected from an online database (https://www.kanjidatabase.com/)[23]. During Pavlovian conditioning, participants learned to associate six CS with three US. Each US was associated with two CS. **b** Choice-induced revaluation: after Pavlovian conditioning, participants made choices between $CS_A^+$ versus $CS_A^0$ (Experiments 1 and 5), $CS_A^0$ versus $CS_A^-$ (Experiment 2), $CS_A^0$ versus either $CS_A^-$ or $CS_A^+$ (Experiment 3), and $CS_{80}^+$ versus $CS_{80}^0$ or $CS_{20}^+$ versus $CS_{20}^0$ (Experiment 4), interleaved with lure decisions. **c** Decision probe: following choice-induced revaluation, participants made binary choices between all possible combinations of CS, or $CS_{80}^+$ versus $CS_{20}^+$ and $CS_{80}^0$ versus $CS_{20}^0$ (Experiment 4), to assess preferences. **d** Attentional control task performed during fMRI repetition suppression (Experiment 5). **e–h** Previously chosen CS (blue dots) are selected more often compared to equivalent CS (black dots) in Experiment 1 (**e**, $N = 40$, $Z = 3.98$, $P < 0.001$, Cohen's $U_3 = 0.85$, Wilcoxon signed-rank test, one-tailed), Experiment 2 (**f**, $N = 40$, $Z = 2.20$, $P = 0.014$, $U_3 = 0.68$, Wilcoxon signed-rank test, one-tailed), Experiment 5 (**h**, $N = 42$, $Z = 3.03$, $P = 0.001$, $U_3 = 0.76$, Wilcoxon signed-rank test, one-tailed) and previously unchosen CS (yellow dots) are selected less often compared to equivalent CS (black dots) in Experiment 1 (**e**, $N = 40$, $Z = 1.97$, $P = 0.025$, $U_3 = 0.70$, Wilcoxon signed-rank test, one-tailed) and Experiment 2 (**f**, $N = 40$, $Z = 1.91$, $P = 0.028$, $U_3 = 0.66$, Wilcoxon signed-rank test, one-tailed) during decision probe. The effect is not present in Experiment 3 (**g**, $N = 44$, $Z = 0.41$, $P = 0.680$, $U_3 = 0.55$, Wilcoxon signed-rank test, two-tailed), indicating that the roughly equal proportion of choices and non-choices of $CS_A^0$ during revaluation had canceled each other out. **i** Behavioral control experiment (Experiment 4), orthogonalizing contributions of go and no-go tagging and associative strength between CS and US to choice probabilities. Previously chosen (go tag) and strongly associated $CS_{80}^+$ is preferred over previously chosen and weakly associated $CS_{20}^+$ (blue dots, $N = 40$, $Z = 3.55$, $P < 0.001$, $U3_1 = 0.75$, $1-\beta > 0.99$, one-sample Wilcoxon signed-rank test, one-tailed), while there is only descriptive evidence for preference of previously unchosen (no-go tag) and strongly associated $CS_{80}^0$ over previously unchosen and weakly associated $CS_{20}^0$ (yellow dots, $N = 40$, $Z = 0.61$, $P = 0.271$, $U3_1 = 0.61$, $1-\beta = 0.23$, one-sample Wilcoxon signed-rank test, one-tailed). Box plot center lines represent sample medians and box bottom/top edges show 25th/75th percentile of the data, respectively. Source data are provided as a Source Data file.

$P < 0.001$, Cohen's $U_3 = 0.85$, Wilcoxon signed-rank test, one-tailed). This effect is mainly driven by preference for $CS_A^+$ in pairwise within-category choice trials between $CS_A^+$ and $CS_B^+$ ($Z = 3.43$, $P < 0.001$, Cohen's $U3_1 = 0.69$, one-sample Wilcoxon signed-rank test vs. 0.5, one-tailed; Supplementary Fig. 2e). Conversely, participants selected $CS_A^0$, the previously non-selected stimulus, compared to $CS_B^0$ less likely ($Z = 1.97$, $P = 0.025$, $U_3 = 0.70$, Wilcoxon signed-rank test, one-tailed). Again, this effect was mainly driven by reduced choice of $CS_A^0$ in pairwise within-category choice trials contrasting $CS_A^0$ and $CS_B^0$ ($Z = 2.05$, $P = 0.020$, $U3_1 = 0.68$, one-sample Wilcoxon signed-rank test vs. 0.5, one-tailed; Supplementary Fig. 2e). The observed dissociation in choice behavior was also evident in a significant interaction effect of CS value × CS type (A or B): $F_{2, 78} = 10.01$, $P < 0.001$, $\eta_p^2 = 0.20$, $1-\beta > 0.99$ (rmANOVA, Fig. 1e). Thus, compared to equivalent CS, participants exhibited a systematic preference for CS they had previously chosen, whereas they

displayed a diminished preference of CS they had previously not chosen.

After having established that an intermediate value CS ($CS_A^0$) could be devalued by non-choices, we next asked in Experiment 2, whether we could induce the exact opposite — increased preference for $CS_A^0$. Therefore, participants ($N = 40$) were presented with decisions between intermediate-value $CS_A^0$ and low-value $CS_A^-$ during the choice-induced revaluation phase. Conceptually replicating the results of Experiment 1, participants in Experiment 2 favored the previously chosen $CS_A^0$ over $CS_B^0$ ($Z = 2.20$, $P = 0.014$, $U_3 = 0.68$, Wilcoxon signed-rank test, one-tailed) during the decision probe, resulting from preference for $CS_A^0$ in pairwise within-category choice trials between $CS_A^0$ and $CS_B^0$ ($Z = 1.93$, $P = 0.027$, $U3_1 = 0.68$, one-sample Wilcoxon signed-rank test vs. 0.5, one-tailed; Supplementary Fig. 2f). Conversely, participants neglected $CS_A^-$ compared to $CS_B^-$ ($Z = 1.91$, $P = 0.028$, $U_3 = 0.66$, Wilcoxon signed-rank test, one-tailed),

resulting from descriptively reduced preference for $CS_A^-$ in pairwise within-category choice trials between $CS_A^-$ and $CS_B^-$ ($Z = 1.41$, $P = 0.079$, $U3_1 = 0.63$, one-sample Wilcoxon signed-rank test vs. 0.5, one-tailed; Supplementary Fig. 2f). Again, there was a significant interaction effect of CS value × CS type ($F_{2, 78} = 4.84$, $P = 0.010$, $\eta_p^2 = 0.11$, $1-\beta > 0.99$, rmANOVA, Fig. 1f) indicating clearly dissociable choice behavior during the decision probe. This pattern of results suggests a value-independent mechanism of choice-induced revaluation.

These results so far show that choices and non-choices act in opposite directions. Consequently, we predicted that choice-induced preference increases and devaluation of $CS_A^0$ would cancel out. To test this prediction, in Experiment 3 ($N = 44$), we presented an equal number of binary decisions between $CS_A^0$ and $CS_A^-$ as between $CS_A^0$ and $CS_A^+$ during choice-induced revaluation. Since choice-induced revaluation effects for $CS_A^0$ should cancel each other out, we did not have directed hypotheses for the choice effects. We thus used two-tailed post hoc tests. As expected, there was no evidence for change in preference for $CS_A^0$ compared to $CS_B^0$ ($Z = 0.41$, $P = 0.680$, $U3 = 0.55$, Wilcoxon signed-rank test, two-tailed, Fig. 1g). Consistently, there was no evidence for $CS_A^0$ preference changes in pairwise within-category choices between $CS_A^0$ and $CS_B^0$ ($Z = 0.12$, $P = 0.905$, $U3_1 = 0.57$, one-sample Wilcoxon signed-rank test vs. 0.5, two-tailed; Supplementary Fig. 2g), suggesting that the effects of choices and non-choices had indeed canceled each other out (interaction effect of CS value × CS type: $F_{2, 86} = 1.31$, $P = 0.280$, $\eta_p^2 = 0.03$, $1-\beta = 0.71$, rmANOVA).

Importantly, the dissociations observed in choice behavior in Experiments 1 and 2 rule out alternative accounts for explaining choice behavior, such as extinction or mere exposure effects. Both accounts would predict unidirectional preference changes for the CS presented during choice-induced revaluation, independent of the choices made (decreases or increases in preference, respectively), which is incompatible with the present results.

However, an alternative explanation for the observed choice pattern is that participants learned simple choice rules for the two CS presented during the revaluation phase, akin to go tags for chosen CS ("choose this stimulus") and no-go tags for unchosen CS ("do not choose this stimulus"). Accordingly, the observed changes in preferences could be attributed to repeating such choice heuristics acquired during revaluation. An additional behavioral experiment (Experiment 4, $N = 40$) was specifically designed to address this possibility. We orthogonalized contributions of associative strength and choice rule by letting participants assign choice-induced go tags to two chosen $CS^+$ that differed in their associative strength to $US^+$ (80% vs. 20% association) and no-go tags to two unchosen $CS^0$ that likewise differed in their associative strength to $US^0$ (80% vs. 20%). According to our hypothesis (associative account), probe phase decisions are guided by the learned associations and the strengthening/weakening of this association during revaluation. Therefore, we expected significantly increased CP for both highly associated stimuli: $CS_{80}^0$ should be preferred over $CS_{20}^0$, and $CS_{80}^+$ should be preferred over $CS_{20}^+$. Contrarily, if choice behavior was instead exclusively driven by learned go and no-go tags (heuristic account), both same-value pairwise CP should be at chance level (CP = 0.50). Due to the directionality of our hypothesis, we used one-tailed tests.

Importantly, there was no significant difference between revaluation CP of $CS_{80}^+$ versus $CS_{80}^0$ and $CS_{20}^+$ versus $CS_{20}^0$ ($Z = 1.19$, $P = 0.234$, $U3 = 0.53$, Wilcoxon signed-rank test, two-tailed), ruling out unequal assignment of go and no-go tags across CS pairs. We observed that participants favored the

previously chosen and strongly associated $CS_{80}^+$ over the previously chosen and weakly associated $CS_{20}^+$ ($Z = 3.55$, $P < 0.001$, $U3_1 = 0.75$, $1-\beta > 0.99$, one-sample Wilcoxon signed-rank test, one-tailed). Descriptively, participants also tended to favor the previously unchosen and strongly associated $CS_{80}^0$ over the previously unchosen and weakly associated $CS_{20}^0$ ($Z = 0.61$, $P = 0.271$, $U3_1 = 0.61$, $1-\beta = 0.23$, one-sample Wilcoxon signed-rank test, one-tailed, Fig. 1i) during the decision probe phase. This pattern of results favors an explanation based on associative strengthening of the memory trace between $CS^+$ and $US^+$, rather than on merely expressing a go tag. However, there is no definite evidence against the alternative explanation that participants learned a no-go tag for the unchosen stimuli. This asymmetric expression of response tendencies might result from differential acquisition of go and no-go choice rules, akin to well-described Pavlovian biases[25]. Presumably, during high-value ($CS_{80}^+$ vs. $CS_{20}^+$) choices, most participants used the learned CS–US associative strength instead of go response tendencies to guide their decisions, while this only tended to be the case for intermediate-value ($CS_{80}^0$ vs. $CS_{20}^0$) choices (Median CP = 0.60). Consistent with modeling and empirical evidence for asymmetric action and inaction learning[26], reverting the initially learned action tendency for $CS_{20}^+$ could have less of an impeding effect on re-acquisition of a no-go response during decision probe, than re-acquisition of a go response for $CS_{80}^0$, which was initially learned with an inaction choice rule.

For each experiment (Experiments 1, 2, 3, and 5), we compared six reinforcement learning models[27] that implemented different ways by which participants could have learned CS–US associations—and updated associative strength during choice-induced revaluation based on fictive reward prediction errors (RPE). The fictive RPE were based on our reasoning that presentation of CS during revaluation would lead to retrieval of the associated US and strengthening/weakening of the chosen/unchosen CS–US association, respectively. Our behavioral results were best captured by a model that differentially updated the learned CS–US associative strengths using fictive RPE elicited by revaluation phase decisions (see Methods section and Supplementary Table 1). Simulations using the best-fitting parameters successfully reproduced the observed empirical choice pattern (with the exception of the observed reduced CP of $CS_B^0$ in Experiment 5, Supplementary Fig. 1e–h). Thus, the best fitting models likely incorporate candidate computational mechanisms underlying the observed choice biases.

**Choices modify univariate neural measures of stimulus-outcome associations.** Having established and replicated the behavioral effect of choice-induced revaluation in three independent behavioral samples, we next tested whether decisions induce changes of neural representations of CS–US associations. In Experiment 5, we used fMRI and leveraged RS effects[10,11,19–21] to measure CS–US associative strength[10,14]. When a neural ensemble is activated twice in brief succession (e.g. by rapid sequential presentation of the same visual stimulus), the second stimulus causes a diminished response. Accordingly, after learning the association between CS and US, the CS should elicit a representation of its associated US. Thus, presentation of the US itself, following the CS, should induce a diminished neural response. If the association between CS and US has been weakened by non-choices during revaluation, the CS is no longer capable of evoking the US representation to the same degree and should therefore elicit a stronger response (less RS). The same logic in reverse applies when the association has been strengthened by choices during revaluation.

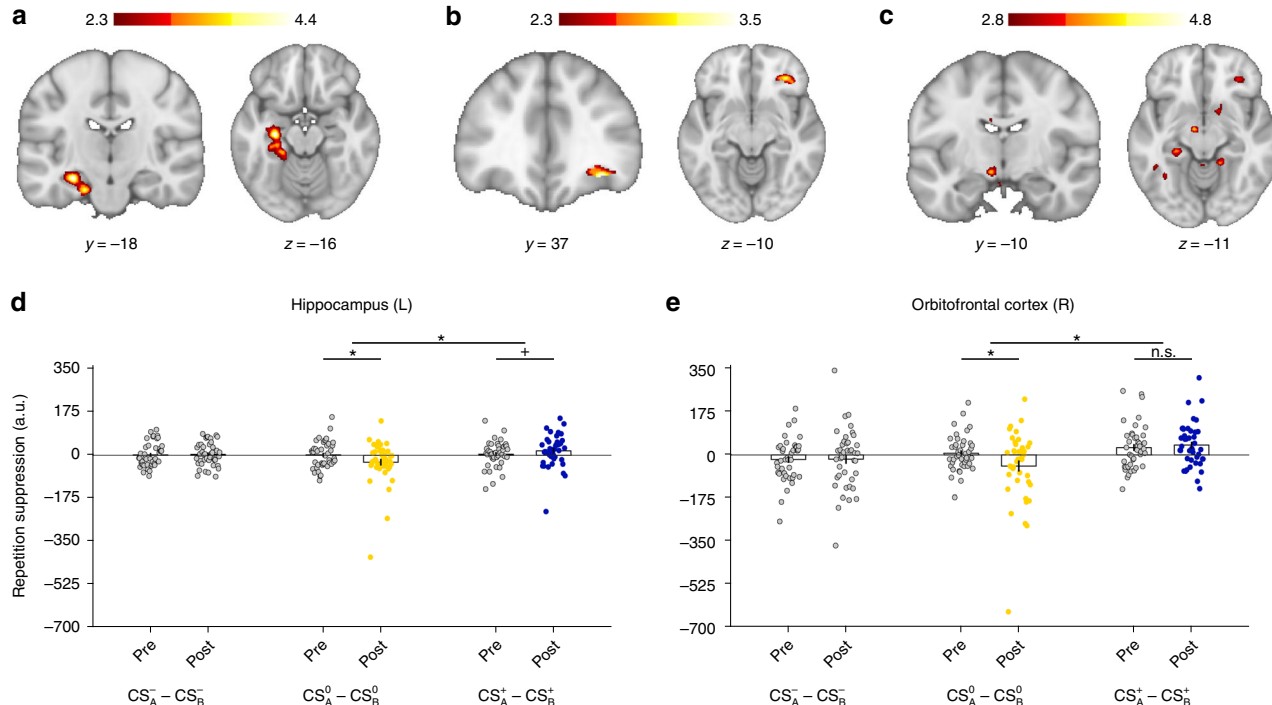

**Fig. 2 Whole-brain effects of choice-induced revaluation. a** Left hippocampus encodes conjunction of decreased repetition suppression for $CS_A^0 - US^0$ relative to $CS_B^0 - US^0$, controlling for activation elicited by $CS_A^0$ and $CS_B^0$ followed by both incorrect outcomes ($US^-$ and $US^+$) (Eq. (1)) AND increased $CS_A^+ - US^+$ repetition suppression relative to $CS_B^+ - US^+$, controlling for activation elicited by $CS_A^+$ and $CS_B^+$ followed by both incorrect outcomes ($US^-$ and $US^0$) (Eq. (2)). **b** Using small-volume correction (independent mask from ref. [28]), the same effect was found in the right lateral orbitofrontal cortex (lOFC). **c** At a lenient threshold of $Z > 2.8$ (uncorrected) we observed the same effect in left ventral tegmental area and right nucleus accumbens. **d** Extracted parameter estimates of the effect in left hippocampus ($N = 42$), showing an interaction effect of CS value and time (PRE or POST), $F_{1, 41} = 4.51$, $P = 0.040$, $\eta_p^2 = 0.10$, $1-\beta = 0.99$, repeated-measures ANOVA (rmANOVA). There was evidence for significant PRE–POST reduction in repetition suppression for $CS_A^0$ ($Z = 1.84$, $P = 0.033$, $U3 = 0.69$, Wilcoxon signed-rank test, one-tailed). However, PRE–POST increase in repetition suppression for $CS_A^+$ was not significant ($Z = 1.48$, $P = 0.070$, $U3 = 0.62$, Wilcoxon signed-rank test, one-tailed). We observed decreased repetition suppression for $CS_A^0$ ($Z = 2.26$, $P = 0.012$, $U3_1 = 0.67$, one-sample Wilcoxon signed-rank test, one-tailed) and increased repetition suppression for $CS_A^+$ ($Z = 2.26$, $P = 0.012$, $U3_1 = 0.69$, one-sample Wilcoxon signed-rank test, one-tailed) during POST, without any evidence for non-zero differences in PRE (all $Zs < 1.05$, $Ps > 0.296$, $U3_1 < 0.62$, one-sample Wilcoxon signed-rank tests, two-tailed) or for the control contrast of either $CS_A^-$ or $CS_B^-$ (all $Zs < 0.31$, $Ps > 0.760$, $U3_1 < 0.55$, one-sample Wilcoxon signed-rank tests, two-tailed). **e** Extracted parameter estimates of the effect in right lOFC ($N = 42$). The interaction effect of CS value and time (PRE or POST) was significant, $F_{1, 41} = 5.57$, $P = 0.023$, $\eta_p^2 = 0.12$, $1-\beta = 0.99$, rmANOVA. We found a significant PRE–POST reduction in repetition suppression for $CS_A^0$ ($Z = 1.77$, $P = 0.039$, $U3 = 0.55$, Wilcoxon signed-rank test, one-tailed), but no evidence of a PRE–POST increase in repetition suppression for $CS_A^+$ ($Z = 0.65$, $P = 0.260$, $U3 = 0.50$, Wilcoxon signed-rank test, one-tailed). However, we found decreased repetition suppression for $CS_A^0$ ($Z = 1.74$, $P = 0.040$, $U3_1 = 0.55$, Wilcoxon signed-rank test, one-tailed) and increased repetition suppression for $CS_A^+$ ($Z = 2.68$, $P = 0.004$, $U3_1 = 0.67$, Wilcoxon signed-rank test, one-tailed) during POST, without evidence for non-zero differences in PRE (all $Zs < 1.77$, $Ps > 0.077$, $U3_1 < 0.62$, Wilcoxon signed-rank tests, two-tailed). Bar plots represent sample means. Error bars indicate standard errors of the means. Asterisks indicate $P$-values $< 0.05$, plus signs represent $P$-values $> 0.05$ and $< 0.10$. Color bars indicate $Z$-values. Source data are provided as a Source Data file.

As in the first three behavioral experiments, participants first learned the six CS–US associations during Pavlovian learning. Following Pavlovian learning, we administered two RS blocks, one immediately before (PRE) and one immediately after (POST) the choice-induced revaluation phase, where participants ($N = 42$) made binary choices between $CS_A^0$ and $CS_A^+$. For the RS effects during POST we had directed hypotheses (increased RS for the previously chosen, and decreased RS for the previously unchosen CS). Therefore, we used one-tailed post hoc tests. For the PRE phase, as well as for the control RS effects of $CS_A^-$ relative to $CS_B^-$, there was no such directed hypothesis and we used two-tailed tests accordingly. Consistent with our hypothesis, we observed both a decrease in RS for $CS_A^0 - US^0$ relative to $CS_B^0 - US^0$, and an increase in RS between $CS_A^+ - US^+$ compared to $CS_B^+ - US^+$ during POST but not during PRE ($Z = 2.53$, $P = 0.006$, $U3 = 0.67$, Wilcoxon signed-rank test, one-tailed) in the left hippocampus (Fig. 2a). Detailed analyses showed that this effect was driven by dissociable effects (interaction effect CS value and time (PRE or

POST), $F_{1, 41} = 4.51$, $P = 0.040$, $\eta_p^2 = 0.10$, $1-\beta = 0.99$, rmA-NOVA): we found decreased RS for $CS_A^0$ ($Z = 2.26$, $P = 0.012$, $U3_1 = 0.67$, one-sample Wilcoxon signed-rank test, one-tailed) and increased RS for $CS_A^+$ ($Z = 2.26$, $P = 0.012$, $U3_1 = 0.69$, one-sample Wilcoxon signed-rank test, one-tailed) during POST, without any evidence for non-zero differences in PRE (all $Zs < 1.05$, $Ps > 0.296$, $U3_1 < 0.62$, one-sample Wilcoxon signed-rank tests, two-tailed) or for the control contrast of either $CS_A^-$ or $CS_B^-$ (all $Zs < 0.31$, $Ps > 0.760$, $U3_1 < 0.55$, one-sample Wilcoxon signed-rank tests, two-tailed, Fig. 2d). These effects arose from (numerically) reduced RS elicited by $CS_A^0$ and increased RS for $CS_A^+$ in separate analyses for $CS_A^0 - US^0$ and $CS_A^+ - US^+$ (Supplementary Fig. 3, Supplementary Notes 2 and 3). Thus, CS choices during choice-induced revaluation increased, whereas non-choices decreased hippocampal CS–US associative strength. However, while there was evidence for significant PRE–POST reduction in RS for $CS_A^0$ ($Z = 1.84$, $P = 0.033$, $U3 = 0.69$, Wilcoxon signed-rank test, one-tailed), the PRE–POST increase

in RS for $CS_A^+$ was not significant ($Z = 1.48$, $P = 0.070$, $U_3 = 0.62$, Wilcoxon signed-rank test, one-tailed).

Furthermore, we found decreased RS for $CS_A^0 - US^0$ relative to $CS_B^0 - US^0$ and increased RS for $CS_A^+ - US^+$ relative to $CS_B^+ - US^+$ in the right lateral orbitofrontal cortex that survived small-volume correction (Fig. 2b). Extraction of parameter estimates from this cluster using an independent region of interest[28] revealed that the interaction effect (CS value and time (PRE or POST), $F_{1, 41} = 5.57$, $P = 0.023$, $\eta_p^2 = 0.12$, $1-\beta = 0.99$, rmANOVA) was driven by a significant PRE–POST reduction in RS for $CS_A^0$ (Fig. 2e, $Z = 1.77$, $P = 0.039$, $U_3 = 0.55$, Wilcoxon signed-rank test, one-tailed) but not by a PRE–POST increase in RS $CS_A^+$ ($Z = 0.65$, $P = 0.260$, $U_3 = 0.50$, Wilcoxon signed-rank test, one-tailed). However, we found decreased RS for $CS_A^0$ ($Z = 1.74$, $P = 0.040$, $U3_1 = 0.55$, Wilcoxon signed-rank test, one-tailed) and increased RS for $CS_A^+$ ($Z = 2.68$, $P = 0.004$, $U3_1 = 0.67$, Wilcoxon signed-rank test, one-tailed) during POST, without evidence for non-zero RS in PRE (all $Zs < 1.77$, $Ps > 0.077$, $U3_1 < 0.62$, Wilcoxon signed-rank tests, two-tailed).

Exploratory analyses at a lenient, uncorrected threshold ($Z > 2.8$, uncorrected) yielded clusters in the left ventral tegmental area (VTA) and the right nucleus accumbens (NAcc, Fig. 2c). Both effects were driven by significantly reduced RS for $CS_A^0$ (VTA: $Z = 1.81$, $P = 0.035$, $U_3 = 0.79$; NAcc: $Z = 2.38$, $P = 0.009$, $U_3 = 0.64$, Wilcoxon signed-rank tests, one-tailed), but only NAcc showed evidence of increased RS for $CS_A^+$ (VTA: $Z = 0.41$, $P = 0.340$, $U_3 = 0.55$; NAcc: $Z = 1.53$, $P = 0.064$, $U_3 = 0.64$, Wilcoxon signed-rank tests, one-tailed). Overall, these RS results suggest that decisions during choice-induced revaluation had clearly dissociable effects on the neural representation of previously learned CS–US associations: While the previously chosen CS exhibited increased associative strength to its related US, the exact opposite effect was true for the previously unchosen CS. Importantly, the observed dissociation of choice-induced increase of RS effects for $CS_A^+$ and decrease of RS effects for $CS_A^0$ and the absence of PRE–POST differences of RS effects for the $CS_A^-$ relative to $CS_B^-$ cannot be explained by general extinction effects resulting from exposition to CS–US associations other than the initially learned associations. Extinction would imply equidirectional PRE–POST changes of all CS–US associations, which is incompatible with the observed results.

**Choice-induced decrease of multivariate neural pattern similarity.** Complementary to the mass-univariate RS-based approach, we performed multivariate fMRI analyses, employing a neural pattern similarity analysis[22] in the left hippocampus and right lateral OFC. Using the same logic as for the RS-based analyses, we reasoned that presentation of a CS would activate neural ensembles representing the associated US. This mnemonic pre-activation should not only be present in trials where the CS was followed by the originally learned US, but also in trials where the CS was followed by any of the other two possible, but not associatively linked US. Similarity of neural patterns related to two CS from the same value category could thus be indicative of associative memory retrieval of a US representation. According to the idea of choice-induced weakening of $CS_A^0$ association with $US^0$ and strengthening of $CS_A^+$ association with $US^+$, our hypothesis therefore was that neural pattern similarity between same-value stimulus–outcome pairs ($CS_A^0 - US^-/CS_B^0 - US^-$ and $CS_A^0 - US^+/CS_B^0 - US^+$; $CS_A^+ - US^-/CS_B^+ - US^-$ and $CS_A^+ - US^0/CS_B^+ - US^0$) should decrease from PRE to POST, indicating less similarity between patterns of interest (i.e. the weakened/strengthened CS and the respective same-value CS, see Methods section for a detailed description). Therefore, we used

one-tailed tests accordingly. For the pairs of control stimuli ($CS_A^- - US^0/CS_B^- - US^0$ and $CS_A^- - US^+/CS_B^- - US^+$), we did not expect changes in neural pattern similarity and thus employed two-tailed tests.

In the left hippocampus ROI, we observed negative PRE–POST change in neural pattern similarity when averaging across all patterns of interest ($t_{41} = 2.09$, $P = 0.021$, $U3_1 = 0.64$, $1-\beta = 0.63$, one-sample $t$-test, one-tailed) and for $CS_A^+/CS_B^+$ pairs ($t_{41} = 1.81$, $P = 0.039$, $U3_1 = 0.57$, $1-\beta = 0.53$, one-sample $t$-test, one-tailed), but only a numerically decreased neural pattern similarity from PRE to POST for $CS_A^0/CS_B^0$ pairs ($t_{41} = 1.01$, $P = 0.144$, $U3_1 = 0.50$, $1-\beta = 0.28$, one-sample $t$-test, one-tailed). Importantly, change of neural pattern similarity for the control stimulus pairs $CS_A^-/CS_B^-$ was positive and not significant ($t_{41} = 0.76$, $P = 0.451$, $U3_1 = 0.57$, $1-\beta = 0.12$, one-sample $t$-test, two-tailed, Fig. 3a).

In the right lOFC ROI, we observed qualitatively similar results as in the left hippocampus: There was significant negative PRE–POST change in neural pattern similarity when averaging across all patterns of interest ($t_{41} = 1.70$, $P = 0.049$, $U3_1 = 0.62$, $1-\beta = 0.40$, one-sample $t$-test, one-tailed). However, there was no evidence of significant change in pattern similarity for $CS_A^+/CS_B^+$ pairs ($t_{41} = 1.23$, $P = 0.113$, $U3_1 = 0.64$, $1-\beta = 0.23$, one-sample $t$-test, one-tailed). There was also no evidence for decreased neural pattern similarity from PRE to POST for $CS_A^0/CS_B^0$ pairs ($t_{41} = 1.55$, $P = 0.061$, $U3_1 = 0.62$, $1-\beta = 0.13$, one-sample $t$-test, one-tailed). Only descriptively, both $CS_A^+/CS_B^+$ pairs and $CS_A^0/CS_B^0$ pairs became less similar from PRE to POST. The change of neural pattern similarity for the control stimulus pairs $CS_A^-/CS_B^-$ was positive, but not significantly different from 0 ($t_{41} = 0.04$, $P = 0.969$, $U3_1 = 0.43$, $1-\beta = 0.05$, one-sample $t$-test, two-tailed, Fig. 3b).

Taken together, these multivariate results conceptually confirm the findings from the mass-univariate RS-based analyses and further support the interpretation that the observed choice effects could be explained by choice-induced changes of associative strength. However, these results should be interpreted with caution, as power was generally low ($1-\beta < 0.80$), most likely resulting from the reduced number of trials included in the analyses (40 trials per CS pair in PRE and POST). Additionally, unlike our RS-based results, changes in neural pattern similarity do not allow to infer the directionality of the effects (i.e. patterns may become more dissimilar both due to strengthening or weakening of the associative trace). Nevertheless, the results from both sets of analyses provide convergent evidence for our hypothesis that revaluation choices changed the degree to which neural US representations were pre-activated by their associated CS.

**Hippocampal CS–US RS correlates with future choices.** We next investigated whether the observed choice-induced modifications of hippocampal CS–US RS were correlated with choice biases during the probe phase. As in Experiments 1 and 2, we had directed hypotheses for the choice effects and thus used one-tailed post-hoc tests.

Unlike in Experiment 1, $CS_A^0$ (unchosen stimulus during revaluation) was not chosen less likely than $CS_B^0$ in the decision probe ($Z = 1.01$, $P = 0.844$, $U_3 = 0.55$, Wilcoxon signed-rank test, one-tailed, Fig. 1h) in Experiment 5. Relatedly, there was no evidence for preference differences in within-category choice trials directly comparing $CS_A^0$ and $CS_B^0$ ($Z = 1.07$, $P = 0.857$, $U3_1 = 0.62$, one-sample Wilcoxon signed-rank test vs. 0.5, one-tailed; Supplementary Fig. 2h). To assess PRE–POST changes of associative strength, participants had to be re-exposed to the initially learned CS–US associations and were explicitly instructed to judge whether the presented CS–US associations were correct.

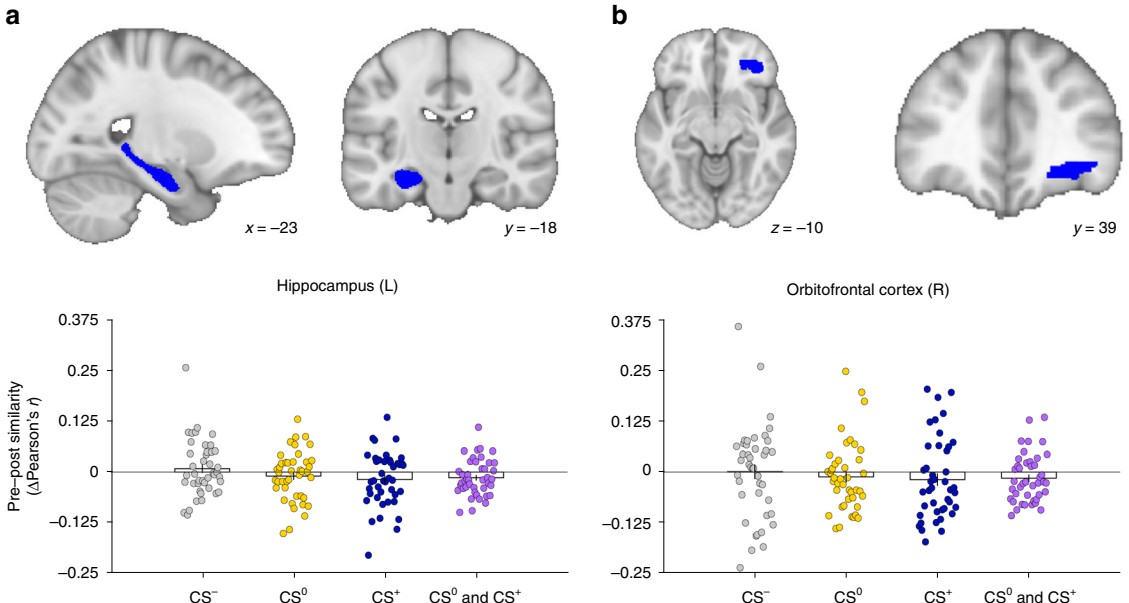

**Fig. 3 Multivariate neural pattern similarity analysis results. a** Multivariate neural pattern similarity analyses in the left hippocampus and **b** right lateral OFC, showing that neural similarity (POST $r$ – PRE $r$, $\Delta$Pearson's $r$) between same value stimulus–outcome pairs decreased from PRE to POST ($N = 42$). $CS^0$ (yellow): $CS_A^0 - US^-/CS_B^0 - US^-$ and $CS_A^0 - US^+/CS_B^0 - US^+$ (Hippocampus: $t_{41} = 1.01$, $P = 0.144$, $U3_1 = 0.50$, $1-\beta = 0.28$, one-sample $t$-test, one-tailed; lateral OFC: $t_{41} = 1.55$, $P = 0.061$, $U3_1 = 0.62$, $1-\beta = 0.13$, one-sample $t$-test, one-tailed); $CS^+$ (blue): $CS_A^+ - US^-/CS_B^+ - US^-$ and $CS_A^+ - US^0/CS_B^+ - US^0$ (Hippocampus: $t_{41} = 1.81$, $P = 0.039$, $U3_1 = 0.57$, $1-\beta = 0.53$, one-sample $t$-test, one-tailed; lateral OFC: $t_{41} = 1.23$, $P = 0.113$, $U3_1 = 0.64$, $1-\beta = 0.23$, one-sample $t$-test, one-tailed); $CS^0$ and $CS^+$ (purple): average of $CS^0$ and $CS^+$ (Hippocampus: $t_{41} = 2.09$, $P = 0.021$, $U3_1 = 0.64$, $1-\beta = 0.63$, one-sample $t$-test, one-tailed; lateral OFC: $t_{41} = 1.70$, $P = 0.049$, $U3_1 = 0.62$, $1-\beta = 0.40$, one-sample $t$-test, one-tailed). Importantly, change of neural pattern similarity for the control stimulus pairs $CS_A^-/CS_B^-$ ($CS^-$, gray) was numerically positive (Hippocampus: $t_{41} = 0.76$, $P = 0.451$, $U3_1 = 0.57$, $1-\beta = 0.12$, one-sample $t$-test, two-tailed; lateral OFC: $t_{41} = 0.04$, $P = 0.969$, $U3_1 = 0.43$, $1-\beta = 0.05$, one-sample $t$-test, two-tailed), indicating higher similarity from PRE to POST. Bar plots represent sample means. Error bars indicate standard errors of the mean. Source data are provided as a Source Data file.

It is well established that restudying of memorized material reverses retrieval-induced forgetting effects[9,29]. Thus, the observed behavioral null effect for $CS_A^0$ might be due to re-exposure and restudy of the original CS–US association, allowing the weakened association between $CS_A^0 - US^0$ to regain its original associative strength.

However, replicating Experiment 1, we observed a choice-induced increase in preference for $CS_A^+$ over $CS_B^+$ ($Z = 3.03$, $P = 0.001$, $U_3 = 0.76$, Wilcoxon signed-rank test, one-tailed, Fig. 1h). This effect was mainly driven by preference for $CS_A^+$ in within-category choices contrasting $CS_A^+$ and $CS_B^+$ ($Z = 1.93$, $P = 0.027$, $U3_1 = 0.62$, one-sample Wilcoxon signed-rank test vs. 0.5, one-tailed; Supplementary Fig. 2h). We therefore focused on this effect in brain–behavior correlations. We hypothesized a positive linear relationship between the difference between choice probabilities of $CS_A^+$ and $CS_B^+$ and the magnitude of hippocampal RS between $CS_A^+ - US^+$ and $CS_B^+ - US^+$), and thus tested the Spearman correlation coefficient one-tailed. The difference of hippocampal RS between $CS_A^+ - US^+$ and $CS_B^+ - US^+$ was positively correlated with the difference between choice probabilities of $CS_A^+$ and $CS_B^+$ ($\rho_{40} = 0.31$, $P = 0.024$, one-tailed; Fig. 4b). The more hippocampal representations of the $CS_A^+ - US^+$ association had been strengthened by choices during revaluation, the more likely participants were to select $CS_A^+$ compared to its non-revalued partner stimulus $CS_B^+$.

Consistently, in a whole-brain analysis we observed a positive relationship between the difference in choice preference for $CS_A^+$ versus $CS_B^+$ and changes in CS–US RS from PRE to POST in the left posterior hippocampus, extending to occipito-temporal complex (Fig. 4c). A similar whole-brain analysis using only the

choices between $CS_A^+$ and $CS_B^+$ as behavioral covariate revealed areas in the bilateral anterior insula and orbitofrontal cortex (Supplementary Fig. 5b). Neither VTA, NAcc, nor the cluster in the lateral orbitofrontal cortex showed relationships with probe phase behavior.

An alternative explanation of our results is based on cached values, a possibility that we address in the Supplementary Methods, Supplementary Notes 1 and 4, and Supplementary Fig. 4.

## Discussion

Using a carefully designed paradigm, we show that decisions bias future choices, even without participants directly experiencing the outcomes of their decisions. Participants were more likely to select CS they had previously chosen, and less likely to select CS they had not chosen, compared to equivalent CS. At the neural level, we found that choices induced alterations to hippocampal and orbitofrontal representations of stimulus–outcome associations that were correlated with future decisions.

The idea that past decisions bias preferences was put forth decades ago[1,4] and evidence for post-decision revaluation has accumulated since[1–3,6] (see ref. [30] for critical discussion). Here, we present behavioral evidence that reward-predictive CS that were chosen in the past are more likely to be selected during future choices, compared to CS of equal value that were not presented. Conversely, we found decreased preferences for CS that were not chosen in the past, compared to equivalent CS, indicating bidirectional effects of choice-induced revaluation. Most importantly, our behavioral findings are independent of rewards, as participants never experienced the outcomes of their

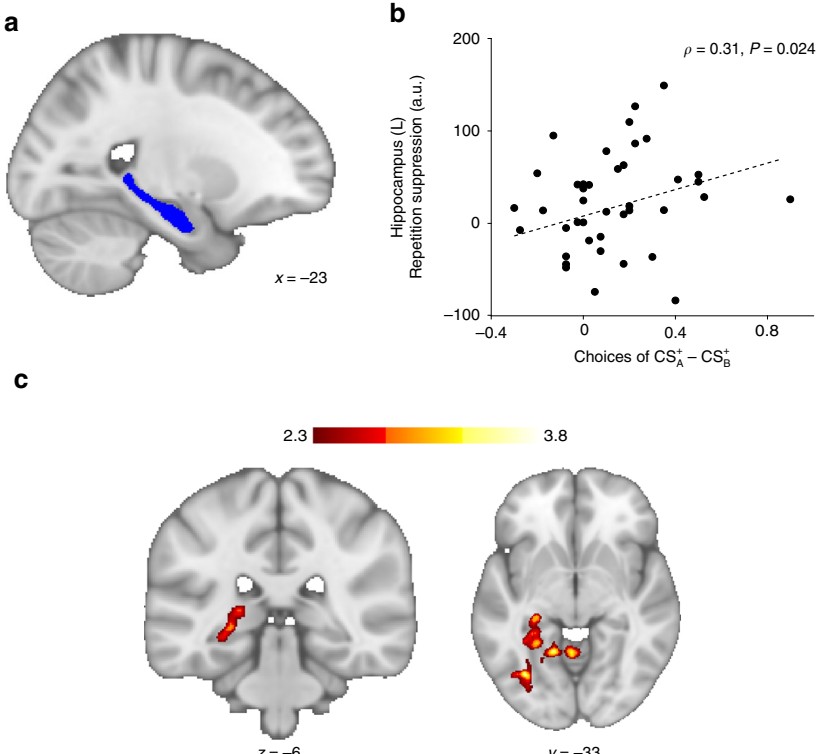

**Fig. 4 Brain–behavior correlation and whole-brain regressions. b** Left hippocampus repetition suppression of $CS_A^+ - US^+$ relative to $CS_B^+ - US^+$, controlling for activation elicited by $CS_A^+$ and $CS_B^+$ followed by both incorrect outcomes ($US^-$ and $US^0$) (Eq. (2)), extracted from an independent anatomical mask (**a**) during POST was positively related to the decision probe difference between overall choice probability of $CS_A^+$ and overall choice probability of $CS_B^+$ ($N = 42$, $\rho_{40} = 0.31$, $P = 0.024$, Spearman correlation, one-tailed). With increasing repetition suppression, participants were more likely to prefer $CS_A^+$ over $CS_B^+$. **c** Whole-brain regression ($N = 42$) showing a positive relationship between the difference of $CS_A^+$ and $CS_B^+$ preference and PRE–POST change of repetition suppression in the left posterior hippocampus. With more positive PRE–POST repetition suppression change, participants are more likely to prefer $CS_A^+$ over $CS_B^+$. Color bar indicates $Z$-values. Source data are provided as a Source Data file.

choices. This suggests that the observed effect could arise from associative memory mechanisms, as also indicated by an additional control experiment designed to rule out the alternative explanation that the observed choice biases could result from learned choice heuristics. Additionally, computational models implementing differentially updated CS–US associative strength of chosen and unchosen options based on fictive prediction errors explained our data best.

Our results are in line with previous reports of choice-induced preference changes[1,3,4,6] and conceptually replicate studies showing changes in stimulus valuation by cued approach training (CAT)[5,31–33]. Similar to the present approach, performance of a button press (go response) upon presentation of go stimuli during CAT induces long-term[31] non-reinforced changes of desirability of go stimuli over no-go stimuli. CAT effects are independent of initial value of the stimuli and rely on integrity[33] and activation[5,31] of ventromedial prefrontal cortex, and interactions between orbitofrontal cortex and ventral striatum[31]. Importantly, the results of Experiment 4 suggest that choice-induced revaluation effects, at least for previously chosen options, seem to go beyond a trained action tendency or choice rule (go response), as observed in CAT.

As most previous studies have presented participants directly with the choice outcomes and thereby confounded contributions of memory and choice mechanisms, our study might be the first to provide evidence for a value-independent associative memory mechanism driving choice-induced preference changes. Furthermore, although associative memory dynamics are a likely candidate mechanism of choice-induced preference changes, the exact

neural mechanisms underlying this phenomenon have remained unknown. Here, we provide evidence that past choices bias future decision-making, in part by modifying the strength of neural stimulus–outcome associations. The present results suggest that, during decision-making, reactivation of stimulus–outcome associations[10,11,14,34], and making a choice renders them subject to nonmonotonic plasticity[16], with the association of the chosen stimulus being strengthened, and the association of the unchosen stimulus being diminished. Since both chosen and unchosen CS activate neural populations representing the respective-associated outcome[10–15], we reason that the observed opposing decision biases are presumably related to additional choice-induced activation of the chosen CS–US association, and absence of such choice-induced activation of the unchosen CS–US associations. Our results suggest that choices can act as self-generated teaching signals[18,35], dynamically altering stimulus–outcome associations stored in memory. However, it should be noted that we did not observe behavioral evidence for choice-induced weakening of the unchosen CS–US association in Experiment 5. This might be due to re-exposure to the initially learned CS–US associations during the POST fMRI run, which presumably allowed the weakened association between $CS_A^0 - US^0$ to regain its original associative strength, in line with studies showing that restudying of memorized material reverses retrieval-induced forgetting effects[9,29].

Even though our data provide evidence for choice-induced changes to associative strength, the current approach does not allow to dissociate which exact features of the US contribute to RS effects. As US value and identity are inextricably linked in our experiment, the observed effects could be related to changes in

CS-dependent pre-activation of identity-related or value-related features of the US. However, both identity-related and value-related features of the US are learned associatively in the present study and are therefore likely retrieved in an associative fashion to guide choices. Alternatively, the observed change in the hippocampal RS could be due to choice-induced alterations of CS representations per se. However, changes of CS representation cannot explain the specificity of RS signals to presentations of the learnt CS–US associations, and, perhaps more importantly, cannot account for the reversed directionality of RS effects depending on choice during revaluation.

Our results suggest that merely making a choice induces plasticity of associative representations in the hippocampus and lateral OFC. In line with the present results, a fronto-hippocampal network comprising hippocampus and lateral OFC seems to be critically involved in reward-related updating of stimulus–outcome associations[14,28]. Whereas the hippocampus has been found to support relational learning and memory processes, including value spread[34] and factorized replay of event trajectories[36], the lateral OFC has been additionally implicated in the resolution of credit-assignment problems in reinforcement learning[28,37].

Our study has at least three implications for current theories of decision-making. First, it proposes a memory-based account for choice-induced revaluation and, more broadly, choice history bias, two well-known, but still poorly understood decision-making phenomena. We show that choices are not only guided by associative representations of value stored in memory, but that decisions themselves dynamically transform associative memories. This suggests that relational structures constituting decision-makers' cognitive maps[38,39] can be distorted through their very own choice behavior. Second, we provide evidence for involvement of the hippocampus and lateral OFC in maintaining and updating of stimulus–outcome associations. This extends previous findings[10,14,28,37] by describing a functional role of the hippocampus in value-based decision-making that is independent of experienced reward. Such a role may more closely resemble naturalistic decision situations where consequences of choices often unravel at distant future time points, rendering credit assignment challenging[28,37]. Third, we provide a mechanism underlying seemingly irrational choice behavior: Even though participants chose between equivalent options, they were biased to prefer chosen, and to neglect non-selected options. The latter might have important implications for explaining subjective preferences, especially in consumer choice behavior[2] and in understanding why humans tend to make coherent decisions, even in conditions characterized by maladaptive choice behavior, such as substance dependence or obsessive compulsive disorder.

Taken together, both our behavioral and neural results support the key prediction that past choices bias future decision-making, partially by altering hippocampal and orbitofrontal representations of stimulus–outcome associations. Our study provides a memory-based mechanism for choice-induced preference change effects[1,3,4,6]. The present study shows that merely retrieving stimulus–outcome associations and making a choice is sufficient to induce plasticity in reward-predictive associations stored in memory.

## Methods

**Participants.** Participants were recruited from the local student community of the Otto von Guericke University Magdeburg and the Heinrich Heine University Düsseldorf, Germany by public advertisements and via online announcements. Only participants indicating no history of psychiatric or neurological disorder and no regular intake of medication known to interact with the central nervous system were included. Participants in all experiments had normal or corrected-to-normal vision and did not report experience with Japanese kanjis or Chinese characters. All participants provided informed written consent before participation and received

monetary compensation for taking part in the study. The study was approved by the local ethics committee at the medical faculty of the Otto von Guericke University Magdeburg, Germany (February 2, 2018, reference number: 19/18) and conducted in accordance with the Declaration of Helsinki.

Forty-nine young, healthy volunteers (age: $M = 23.93$, $SD = 2.90$ years, 18 males) participated in Experiment 1. Seven participants were excluded from statistical analyses due to lacking engagement in the cover task during the Pavlovian conditioning phase that served as an attentional control (<10% responses in trials that required to indicate the color of the square surrounding the conditioned stimuli. Two additional participants had to be excluded due to not passing the manipulation check (high-value option selected <50% during choice-induced revaluation), thus leaving a total of $N = 40$ participants for final analyses.

Sixty-four young, healthy volunteers (age: $M = 23.47$, $SD = 3.79$ years, 26 males), participated in Experiment 2. Ten participants were excluded from statistical analyses due to lacking engagement in the cover task during the Pavlovian conditioning phase, and 13 subjects were excluded due to not passing the manipulation check (intermediate valued option selected <50% during choice-induced revaluation), one additional participant had to be excluded due to a technical error, leaving $N = 40$ participants for statistical analyses.

Sixty-one young, healthy volunteers (age: $M = 23.26$, $SD = 3.27$ years, 23 males), participated in Experiment 3. Ten participants were excluded from statistical analyses due to lacking engagement in the cover task during the Pavlovian conditioning phase, and six subjects were excluded due to not passing the manipulation check (high-value option selected <50% during choice-induced revaluation), one additional participant had to be excluded due to a technical error (no data was recorded), leaving $N = 44$ participants for statistical analyses.

Fifty-two young, healthy volunteers (age: $M = 22.06$, $SD = 3.69$ years, 20 males), participated in Experiment 4. Twelve subjects were excluded due to not passing the manipulation check (high-value option selected <50% during choice-induced revaluation), leaving $N = 40$ participants for statistical analyses.

Fifty-eight young, healthy and magnetic resonance imaging (MRI)-compatible volunteers (age: $M = 24.61$, $SD = 4.01$ years, 30 males) participated in Experiment 5 (functional MRI (fMRI) experiment). One participant fell asleep during the POST revaluation fMRI-RS run, three participants discontinued the MRI acquisition (one due to claustrophobia, two reported a headache during task performance). Twelve additional subjects were excluded due to not passing the manipulation check (high-value option selected <50% during choice-induced revaluation), leaving $N = 42$ participants for statistical analyses.

**Behavioral task—ratings.** Participants received written instructions for the experiment and were instructed once again on the computer screen. The experiments were programmed in MATLAB 2012b (MATLAB and Statistics Toolbox Release 2012b, The MathWorks, Inc., Natick, MA, USA, v8.0.0.783), using Psychophysics Toolbox[40] (version 3) and MATLAB 2019a (v9.6.0.1072779). Before the task, participants rated 25 different sweet and high-caloric food items selected from an online database[24]. Subjects were instructed to indicate the subjective desirability of the food items by using the "y" and "m" button on a standard German (QWERTZ) computer keyboard to position a red slider bar on a white visual analog scale (VAS) between 0 (not liked) and 100 (very much liked). The lowest- (Experiment 1: $M = 17.5$, $SD = 20.51$; Experiment 2: $M = 16.65$, $SD = 16.29$; Experiment 3: $M = 18.59$, $SD = 22.15$; Experiment 5: $M = 20.41$, $SD = 20.44$), and highest-rated (Experiment 1: $M = 96.55$, $SD = 7.10$; Experiment 2: $M = 94.85$, $SD = 9.08$; Experiment 3: $M = 95.21$, $SD = 14.52$; Experiment 4: $M = 96.63$, $SD = 6.62$; Experiment 5: $M = 97.79$, $SD = 4.96$) food item as well as a food item rated with the median value of all ratings (Experiment 1: $M = 57.43$, $SD = 11.55$; Experiment 2: $M = 56.23$, $SD = 9.80$; Experiment 3: $M = 57.14$, $SD = 13.95$; Experiment 4: $M = 56.48$, $SD = 9.95$; Experiment 5: $M = 60.07$, $SD = 11.71$) were selected for Pavlovian conditioning. We explicitly decided for clearly differentiable pictures of food items in order to elicit activation of differential neural ensembles coding for those stimuli and to facilitate learning of vivid memories of stimulus–outcome associations. Next, subjects rated 20 (11 in Experiment 1) Japanese kanjis[23] according to subjective value/liking by using the "y" and "m" button on a standard computer keyboard to position a red slider bar on a VAS between 0 (not liked) and 100 (very much liked). The six kanjis rated closest to 50 (equivalent to neutral) were selected and their order was randomized before being associated with the food items in Pavlovian conditioning. The subjective values/liking of the six selected kanjis did not differ significantly from each other in Experiments 1, 3, 4, and 5 (main effects of stimulus: all $Fs < 1.14$, $Ps > 0.344$, $\eta_p^2 < 0.03$, rmANOVA). However, there was a main effect of stimulus in Experiment 2 ($F_{5,\ 195} = 2.65$, $P = 0.024$, $\eta_p^2 = 0.064$, rmANOVA), resulting from a significantly higher pre-rating for control stimulus $CS_A^+$ compared to control stimulus $CS_B^+$ ($t_{39} = 3.13$, $P = 0.003$, paired-samples $t$-test). More importantly, both critical pairs ($CS^-$ and $CS^0$) did not differ significantly (all $ts < 0.79$, $Ps > 0.437$, paired-samples $t$-tests).

**Behavioral task—Pavlovian conditioning.** Participants learned to associate the selected kanjis (conditioned stimuli, CS) with differently valued outcomes (unconditioned stimuli, US) by repeatedly observing one CS (2000 ms), followed by an inter-stimulus interval (1000 ms) marked by a fixation cross, and presentation of one US (2000 ms). Each trial was separated by an inter-trial-interval (ITI) marked by a gray screen. The ITI per trial was drawn from a discretized $\gamma$-distribution

(shape = 5, scale = 0.9) truncated for an effective range of values between 3000 and 8000 ms. Across all CS, CS–US couplings were interleaved with 20% CS–no-US couplings. This was intended to create an association that would not rapidly extinguish due to extinction effects related to not being presented with the associated US during the subsequent choice phases. Each US was associated with two different CS, resulting in three pairs of differently valued CS: $CS^+_{A/B}$, $CS^0_{A/B}$ and $CS^-_{A/B}$. Participants completed 180 trials (19 of the participants included in Experiment 1 had performed 240 trials) of Pavlovian conditioning, 30 (40, respectively) trials per CS–US association. In Experiment 4, participants performed 160 trials and learned associations between four neutrally rated CS and an intermediate value outcome and a high value outcome (40 trials per CS–US association), resulting in two pairs of differently valued CS: $CS^+_{80}$, $CS^-_{20}$ and $CS^+_{80}$, $CS^-_{20}$. Importantly, one of the two CS in each pair was followed by the outcome with a probability of 80% ($CS^+_{80}$ and $CS^+_{80}$) while the other CS was followed by the US in 20% ($CS^-_{20}$ and $CS^-_{20}$) of the trials. The participants were presented with a cover story for the experiment: They were told to imagine themselves in preparation for a journey to Japan during which they would need to learn kanjis associated with certain food items to be ready to select their favorite sweets in Japanese shops. As an attentional control, we introduced a simple binary classification task, presenting participants with a red or blue square surrounding the CS. Each CS was equally often presented with a red or blue square and color of the square did not predict US contingency. Subjects were instructed to react as quickly and as correctly as possible by pressing "y" upon seeing a blue square and pressing "m" upon seeing a red square surrounding the CS. Pavlovian conditioning was split up in three blocks of 60 (80, respectively) trials, interleaved with self-paced breaks. In Experiment 5, both ratings and Pavlovian conditioning were performed outside the MRI scanner.

Due to a coding error in the script creating pseudo-randomized reward schedules in Experiment 1, 2, and 3 (which we spotted during setup of Experiment 4), CS were not followed by outcomes in exactly 80% of trials equivalently across all CS. For each experiment, we had set up five different reward schedules, assigning different outcome probabilities to each CS. In Experiment 1, participants received the following outcome probabilities on average: $CS^-_A$ (Probability: $M = 0.80$, range = 0.70–0.87), $CS^-_B$ ($M = 0.78$, range = 0.73–0.88), $CS^0_A$ ($M = 0.78$, range = 0.70–0.87), $CS^0_B$ ($M = 0.78$, range = 0.63–0.90), $CS^+_A$ ($M = 0.80$, range = 0.70–0.90), $CS^+_B$ ($M = 0.86$, range = 0.73–0.93). In Experiment 2, participants received the following outcome probabilities on average: $CS^-_A$ ($M = 0.80$, range = 0.73–0.87), $CS^-_B$ ($M = 0.76$, range = 0.73–0.83), $CS^0_A$ ($M = 0.78$, range = 0.70–0.87), $CS^0_B$ ($M = 0.78$, range = 0.63–0.90), $CS^+_A$ ($M = 0.80$, range = 0.70–0.90), $CS^+_B$ ($M = 0.88$, range = 0.83–0.93). In Experiment 3, participants received the following outcome probabilities on average: $CS^-_A$ ($M = 0.80$, range = 0.73–0.87), $CS^-_B$ ($M = 0.77$, range = 0.73–0.83), $CS^0_A$ ($M = 0.79$, range = 0.70–0.87), $CS^0_B$ ($M = 0.77$, range = 0.63–0.90), $CS^+_A$ ($M = 0.79$, range = 0.70–0.90), $CS^+_B$ ($M = 0.88$, range = 0.83–0.93). It should be noted that these minor differences in outcome probabilities between CS cannot account for the observed choice-induced revaluation effects, as the respective chosen CS on average received less outcomes ($CS^+_A$ in Experiment 1) than or an equal number of outcomes ($CS^0_A$ in Experiment 2) as their same-valued partner stimuli. Contrarily, the respective unchosen CS on average received more outcomes ($CS^-_A$ in Experiment 2) than or an equal number of outcomes ($CS^0_A$ in Experiment 1) as their same-valued partner stimuli. Thus, the outcome probabilities assigned to each CS would have worked against the hypothesized effects. Consistently, there were no significant correlations between the outcome probability during Pavlovian conditioning and decision probe overall or pairwise within-category choice probability ($\rho s < 0.29$, $Ps > 0.067$, Spearman correlations, two-tailed). Importantly, Experiment 5 was not affected from this error, as reward schedules were created with a different script in which we correctly coded that each CS would be followed by an outcome in 80% of the trials.

**Behavioral task—choice-induced revaluation.** After completion of Pavlovian conditioning, participants were presented with repeated choices (28 trials) between a $CS^+_A$ versus a $CS^0_A$ (Experiments 1 and 5), $CS^0_A$ versus a $CS^-_A$ (Experiment 2), a $CS^0_A$ versus either a $CS^+_A$ or a $CS^-_A$ (14 trials) (Experiment 3), or a $CS^+_{80}$ versus a $CS^0_{20}$ and a $CS^+_{20}$ versus a $CS^0_{80}$ (Experiments 4), interleaved with lure decisions (28 trials) between four other neutrally rated kanjis that had never been presented during Pavlovian learning and thus were not associated with any of the US. The choice-induced revaluation phase served as the crucial manipulation in all experiments and was systematically varied across studies. Choice probability (CP) for the high-value CS served as a control for learning and as a manipulation check. Only participants selecting the higher valued CS more than 50% (CP ≥ 0.50) were included in the final analysis, as we reasoned that choice-induced revaluation choices would (1) represent a marker of having learned the true associative values of the CS, (2) be a measure for learning, independent of the actual decision probe phase data (avoiding biased and arbitrary decisions for exclusion of participants), and (3) allow us to exclude decision makers showing random, or arbitrary choice behavior. Choice options were presented for 1500 ms and the chosen option was highlighted by a gray square surrounding the chosen CS. If participants did not respond within the time-window, a time-out message was displayed, and the respective trial was repeated at the end of the choice-induced revaluation phase. Order (left/right) of choice options was counterbalanced to avoid simple

response patterns or decision rules (e.g. "always press left"). Participants were instructed to imagine themselves in a Japanese shop, where they would like to buy their favorite food items based on the previously learned kanjis (CS). Participants were told that one of the choice trials would randomly be drawn and their choice would determine which food item (US) they would receive as a bonus upon completion of the experiment. Participants selected choice options by pressing the "y" (left option) or "m" (right option) button (left or right index finger on an MRI-compatible response box in Experiment 5). Importantly, participants were not presented with the US related to their chosen or unchosen CS to dissociate the observed effects from outcome-related relearning of CS–US associations. We assumed that presentation of a CS would pre-activate neural ensembles coding for the associated US. Consequently, we expected that choosing a CS would induce strengthening of the chosen option's CS–US association, whereas not choosing a CS would weaken the unchosen option's CS–US association.

**Behavioral task—decision probe.** Following choice-induced revaluation, participants were presented with repeated binary choices (120 trials) between all possible CS combinations to assess CS preferences. Every CS combination was presented eight times in pseudo-random order. In Experiment 4, participants made choices between the two same-value pairs of stimuli that were differently strong associated to their respective outcomes ($CS^+_{80}$ versus $CS^-_{20}$ and $CS^+_{80}$ versus $CS^-_{20}$). Here, every CS pair was presented 10 times in pseudo-random order. Choice options were presented for 1500 ms. If participants did not respond within this time-window, a time-out message was displayed, and the respective trial was repeated at the end of the decision probe phase. Participants selected choice options by pressing the "y" (left option) or "m" (right option) button (left or right index finger on MR-compatible response box in Experiment 5). Order (left/right) of choice options was counterbalanced. Participants were instructed that their shopping bag was torn, and they had to return to the shop for buying their favorite food items based on the previously learned kanjis (CS). Again, participants were told that one of the choice trials would randomly be drawn and their choice would determine which food item (US) they would receive as a bonus upon completion of the experiment. Importantly, participants were again not presented with the US related to their chosen or unchosen CS.

**fMRI-RS task (Experiment 5).** After Pavlovian conditioning outside the MRI-scanner, we administered two fMRI-RS blocks, one immediately before (PRE) and one immediately after (POST) the revaluation phase to assess choice-induced effects of fMRI-RS. Every possible combination of CS and US (18 combinations) was presented 20 times each (360 trials in total). In one-third of the trials, the originally learned CS–US associations were presented, the remaining two-third of trials contained incorrect CS–US associations. In every trial, a CS was presented for 700 ms, followed by an interstimulus interval (fixation cross) for 400 ms and a US for 700 ms. The intertrial interval was drawn from a discretized γ-distribution (shape = 2.01, scale = 1), truncated for an effective range of values between 2000 and 6000 ms. Order of trials was pseudo-random, between-trial repetition of CS or US did not occur. Additionally, every batch of 18 trials contained every possible combination of CS–US association to avoid comparison of temporally distal trials and between-trial biases in RS introduced by, e.g. fluctuations of attention, "novelty" or surprise. During both runs of fMRI-RS, participants performed an attentional control task. After a pseudo-random 20% of trials, participants were presented with probe trials in which they were asked to indicate whether or not the previously seen CS–US-association matched the true CS–US-association learned during Pavlovian conditioning via button presses with their right and left index fingers on an MRI-compatible response box. Correct responses were rewarded with 0.05€ and incorrect responses or time-out trials (without a response by the participant within 2500 ms after onset of the probe trial) resulted in a 0.05€ penalty which would be summed up as a bonus upon completion of the experiment. On average, participants earned a bonus of 5.93€ ($SD = 1.05$). Performance during the attentional control task was generally high (overall probability of correct answers, excluding time-out trials: $M = 0.92$, $SD = 0.06$ ($t_{41} = 41.54$, $P < 0.001$, one-sample $t$-test vs. chance level (0.5)), with no evidence for a difference in performance between PRE and POST choice-induced revaluation run (PRE: $M = 0.92$, $SD = 0.07$; POST: $M = 0.92$, $SD = 0.072$; $t_{41} = 0.06$, $P = 0.95$, paired-samples $t$-test).

**fMRI acquisition.** Two runs of fMRI were recorded with a 3 Tesla Siemens PRISMA MR-system (Siemens, Erlangen, Germany), using a 64-channel head coil. Blood oxygenation level-dependent (BOLD) signals were acquired using a multi-band accelerated T2*-weighted echo-planar imaging (EPI) sequence (multi-band acceleration factor 2, repetition time (TR) = 2000 ms, echo time (TE) = 30 ms, flip angle = 80°, field of view (FoV) = 220 mm, voxel size = 2.2 × 2.2 × 2.2 mm, no gap). Per volume, 66 slices covering the whole brain, tilted by ~15° in z-direction relative to the anterior–posterior commissure plane were acquired in interleaved order. The first five volumes of the functional imaging time series were automatically discarded to allow for T1 saturation. After each run, a $B_0$ magnitude and phase map was acquired to estimate field maps and $B_0$ field distortion during preprocessing (TR = 660 ms, TE 1 = 4.92 ms, TE 2 = 7.38 ms, flip angle = 60°, FoV = 220 mm). Additionally, before the PRE choice-induced revaluation

fMRI-RS run, a high-resolution three-dimensional T1-weighted anatomical map (TR = 2500 ms, TE = 2.82 ms, FoV = 256 mm, flip angle = 7°, voxel size = 1 × 1 × 1 mm, 192 slices) covering the whole brain was obtained using a magnetization-prepared rapid acquisition gradient echo (MPRAGE) sequence. This scan was used as anatomical reference to the EPI data during the registration procedure.

**Behavioral analyses.** Data were analyzed in MATLAB 2012b (v8.0.0.783), 2017a (v9.2.0.556344), and 2019a (v9.6.0.1072779) (The MathWorks, Inc., Natick, MA, USA) using custom analysis scripts. For the manipulation check, indicating learning of the CS–US-associations, average choice probabilities (CPs) for the higher valued CS were calculated (one average in Experiments 1, 2, and 5, two averages in Experiments 3 and 4) by summing up choices of the higher-valued CS and dividing by the number of choice trials with a recorded response. This CP was compared against the inclusion criterion of CP ≥ 0.50. If participants had chosen the high-valued CS in more than 50% (or in exactly 50%) of the trials of choice-induced revaluation, they were included in the final analyses. For the decision probe phase, we computed an average overall CP per CS per subject including all binary decisions in which the respective CS was present (count data, 1 representing selection of the CS, 0 representing selection of the alternative CS). Lacking a non-parametric alternative to the parametric two-way repeated-measures analysis of variances (rmANOVA), distributions of mean overall CPs were analyzed at the group level with a rmANOVA with factors CS valence (3) and stimulus type (2) and post-hoc Wilcoxon sign-rank tests for paired samples, focusing on the pairwise comparison of stimulus types within each level of valence. We hypothesized a main effect of CS valence and an interaction effect of CS valence × CS type (A vs. B), resulting from higher CPs for previously chosen CS relative to the equivalent control CS and lower CPs for previously unchosen CS relative to the equivalent control CS (Experiments 1, 2, and 5). However, we expected absence/no evidence for such an interaction effect in Experiment 3 in which a $CS_A^0$ was chosen and unchosen equally often (resulting in no change of the preference relative to the control CS). Additionally, we performed one-sample Wilcoxon sign-rank tests of overall CP against chance level (CP = 0.50). As an alternative measure of choice preference in addition to overall CP, we also computed pairwise choice probabilities by comparing choice ratios of the two CS within each valence category with one-sample Wilcoxon sign-rank against chance level (CP = 0.50).

Experiment 4 was specifically designed to rule out the possibility that participants did not guide their choices based on (altered) associative strength between CS and US but had simply learned choice rules for the two CS presented during the revaluation phase (go tag for chosen CS, no go tag for unchosen CS). We thus aimed to orthogonalize contributions of associative strength and choice rule, by assigning go tags to the two chosen $CS^+$ that differed in their associative strength to $US^+$ and no-go tags to the two unchosen $CS^0$ that differed in their associative strength to $US^0$. Our hypothesis was that if choice behavior was exclusively driven by these tags participants had learned, go tags for both chosen $CS_{80}^+$ and $CS_{20}^+$ and no-go tags for both unchosen $CS_{80}^0$ and $CS_{20}^0$, there should be no evidence for both same-value pairwise choice probabilities different from chance level (CP = 0.50). However, if the choices were made based on the learned associations and the associative strengthening/weakening of the memory trace between CS and US, there should be a significantly increased choice probability for $CS_{80}^0/CS_{80}^+$ that were more strongly associated with their respective outcomes.

As we had formulated directional hypotheses for the choice effects, we performed one-tailed (post-hoc) tests. In Experiment 3, there were no directional hypotheses for the choice effects, therefore, we used two-tailed post-hoc tests. We report effect sizes $\eta_p^2$ for rmANOVAs, Cohen's $U_3$ for Wilcoxon signed-rank tests and Cohen's $U3_1$ for one-sample Wilcoxon signed-rank tests (range: 0–1, 0.5 indicating no effect), calculated in the MATLAB-based Measures-of-Effect-Size-toolbox[41]. Based on the reported effect sizes $\eta_p^2$ we additionally indicate post-hoc achieved power (1–β) for the hypothesized interaction effects of CS valence × CS type in rmANOVAs across behavioral analyses in Experiments 1, 2, 3, and 5. Based on the means and standard deviations for both choice probabilities, we indicate post-hoc achieved power for Experiment 4. All power analyses were conducted in G*Power[42,43] (v3.1.9.2).

**Univariate fMRI data analysis.** We exploited fMRI-RS effects (rapid, repeated presentation of the same stimulus or pre-activation of a stimulus by associated stimuli elicits reduced neural responses, as stimuli are represented by overlapping neural ensembles[19,20]) to investigate choice-related changes in neural representations of CS–US associations. As conditioning enhances CS's ability to pre-activate neural ensembles coding for US, we expected a change of CS–US associative strength, as measured by RS after choice-induced revaluation. Strong CS–US associations should elicit high fMRI-RS effects (i.e. low activation), whereas weak CS-US associations should elicit low fMRI-RS effects (i.e. high activation). We expected decreased neural representations of CS–US associations for $CS_A^+$ and increased choice-related neural representations of CS–US associations for $CS_A^0$ after choice-induced revaluation.

All univariate fMRI analyses steps were performed using tools from the functional magnetic resonance imaging of the brain (FMRIB) Software Library (FSL, v6.0)[44]. Preprocessing included motion correction using rigid-body registration to the central volume of the functional time series[45], correction for geometric distortions using the field maps and an $n$-dimensional phase-unwrapping algorithm ($B_0$ unwarping)[46], slice timing correction using Hanning windowed sinc interpolation, high-pass filtering using a Gaussian-weighted lines filter of 1/100 Hz. EPI images were registered with the high-resolution brain images and normalized into standard (MNI) space using affine linear registration (boundary-based registration) as well as nonlinear registration[47,48]. Functional data were spatially smoothed using a Gaussian filter with 6 mm full-width at half maximum. We applied a conservative independent components analysis (ICA) to identify and remove obvious artefacts. Independent components were manually classified as signal or noise based on published guidelines[49], and noise components were removed from the functional time series. General linear models (GLMs) were fitted into pre-whitened data space to account for local autocorrelations[50]. The individual level (first level) GLM design matrix per run and participant included 22 box-car regressors in total. Eighteen regressors coded for onsets and durations of all 18 presented CS–US association trials (each modeled as single events of 1800 ms duration), one regressor coded for onsets and durations of the three within-run pauses (each 45 s), one regressor coded for onsets and durations of the attentional control task probes, two regressors coded onsets and durations of left and right button presses (delta stick functions on the recorded time of response button presses) and the six volume-to-volume motion parameters from motion correction during preprocessing were entered. Regressors were convolved with a hemodynamic response function (γ-function, mean lag = 6 s, SD = 3 s). Each first level GLM included five contrasts to estimate individual per run contrasts of parameter estimates for (1) lower $CS_A^0 - US^0$ RS relative to the other equivalent $CS_B^0$, controlling for all other combinations of $CS^0$ presentations (Eq. (1)), (2) higher $CS_A^+ - US^+$ RS relative to the other equivalent $CS_B^+$, controlling for all other combinations of $CS^+$ presentations (Eq. (2)), (3) higher $CS_A^-$ RS relative to the other equivalent $CS_B^-$, controlling for all other combinations of $CS^-$ presentations, (4) Conjunction of (1) and (2), i.e. voxels coding for both decrease of $CS_A^0 - US^0$ RS and increase of $CS_A^+ - US^+$ RS (Eq. (3)), (5) right vs. left button press. Two separate PRE and POST choice-induced revaluation second level (group level) GLMs were carried out by submitting individual level parameter estimates to mixed-effects statistics and ordinary least-squares (OLS) regression for higher-level contrast of parameter estimates (COPE) estimation. To control for multiple comparisons, cluster-based correction with an activation threshold of Z > 2.3 using a cluster-extent threshold of P < 0.05 was applied at the whole-brain level.

The key tests for our hypothesis were focused on the effect of revaluation choices on CS–US-associations during the POST run. The PRE run served as a control to rule out potential baseline differences in RS for $CS_A^0$ or $CS_A^+$.

A priori regions-of-interest (ROIs) comprised the lateral orbitofrontal cortex (lOFC) and the hippocampus, as those regions have been implicated in representation and adaptive changes of stimulus–outcome associations, respectively[10,14,21,28]. We investigated POST choice-induced revaluation conjunction effects (Eq. (3)) to identify regions involved in processing of choice-induced changes to CS–US associations. An independent functional mask of a contrast investigating stimulus–outcome associations from a previous study[28] (restricted along the $z$-direction from –6 to –14 to constrain spatial extent), was used for small-volume correction ($P_{SVC}$) of the bilateral lOFC. The small-volume corrected functional activation mask from the conjunction contrast was used to extract contrast parameter estimates of the $CS_A^0$ contrast and the $CS_A^+$ contrast. Additionally, an independent anatomical mask of the hippocampus (Harvard–Oxford Atlas) was used to extract PRE and POST choice-induced revaluation contrast parameter estimates of the $CS_A^0$ contrast and the $CS_A^+$ contrast. PRE versus POST comparisons of activation were carried out using a repeated-measures ANOVA and post-hoc Wilcoxon-sign rank tests. As we had formulated directed hypotheses, and because parameter estimates were extracted from family-wise error corrected ROIs, we used one-tailed post-hoc tests. We report effect sizes $\eta_p^2$ for rmANOVAs, Cohen's $U_3$ for Wilcoxon signed-rank tests and Cohen's $U3_1$ for one-sample Wilcoxon signed-rank tests (range: 0–1, 0.5 indicating no effect), calculated in the MATLAB-based Measures-of-Effect-Size-toolbox[41]. Based on the reported effect sizes $\eta_p^2$ we additionally indicate post-hoc achieved power (1–β) for the hypothesized interaction effects of CS valence × time in rmANOVAs, calculated with G*Power[42,43] (v3.1.9.2). Additionally, we explored functional activation not surviving whole-brain or small-volume family-wise error corrections by thresholding activation maps at Z = 2.8 and extracting contrast estimates of the $CS_A^0$ contrast and the $CS_A^+$ contrast for brain–behavioral correlations.

Extracted parameter estimates were additionally used for brain–behavioral correlations using non-parametric Spearman correlations. For brain–behavior correlations, we had the directed hypotheses that associative strength, as measured by RS should be positively related to the difference between overall CP of $CS_A^+$ and overall CP $CS_B^+$ and negatively related to difference between overall CP $CS_A^0$ and overall CP $CS_B^0$. Additionally, to refine our insights in the relationship of neural and behavioral results, we also correlated RS with pairwise CP for $CS_A^+$ versus $CS_B^+$, again assuming a positive relationship and pairwise CP for $CS_A^0$ versus $CS_B^0$, predicting a negative relationship. Due to our directed hypotheses, and because parameter estimates were extracted from family-wise error corrected ROIs, we performed one-tailed tests on Spearman correlation coefficients.

Furthermore, whole-brain voxel-wise regressions were applied at the group level for both PRE and POST run and also a group level analysis on the activation change from PRE to POST run. We used demeaned individual behavioral difference regressors of overall CP of $CS_A^+$ – overall CP $CS_B^+$ and overall CP $CS_A^0$ – overall CP $CS_B^0$ and demeaned pairwise within-category CP of trials directly comparing $CS_A^+$ and $CS_B^+$ and $CS_A^0$ and $CS_B^0$, respectively.

**Multivariate fMRI data analysis.** Complementary to the univariate RS-based approach, we performed confirmatory multivariate fMRI analyses, employing a neural pattern similarity analysis (a variant of RSA[22]) in the left hippocampus and right lateral OFC to further support the hypothesized associative strengthening/weakening mechanism. As for the RS-based analyses we assumed that presentation of a CS should induce pre-activation of neural ensembles coding the respective associated US. This mnemonic pre-activation should not only be present in trials where the CS was followed by the originally learned US, but also in trials where the CS was followed by any of the other two possible, but not associatively linked US. Similarity of neural patterns related to two CS from the same value category could thus potentially be indicative of associative memory retrieval of a US representation. According to the idea of choice-induced weakening of the $CS_A^0$ association with $US^0$ and strengthening of the $CS_A^+$ association with $US^+$, our hypothesis was that neural similarity between same stimulus–outcome pairs ($CS_A^0 - US^-/CS_B^0 - US^-$ and $CS_A^0 - US^+/CS_B^0 - US^+$; $CS_A^+ - US^-/CS_B^+ - US^-$ and $CS_A^+ - US^0/CS_B^+ - US^0$) should decrease from PRE to POST, indicating less similarity between patterns of interest (i.e. the weakened/strengthened stimulus and the respective partner stimulus). The assumption of decreased neural pattern similarity for pairs of $CS_A^+$ and $CS_B^+$ as a result of choice-induced strengthening of $CS_A^+$ is based on two grounds. Firstly, we assumed that both $CS_A^+$ and $CS_B^+$ would equivalently, and partially reinstate the pattern representing $US^+$ during PRE, but repeated retrieval of $US^+$ and choice of $CS_A^+$ should selectively strengthen the association between $CS_A^+$ and $US^+$ during POST. Since $CS_B^+$ is not presented during revaluation and thus not actively rehearsed, the memory trace between $CS_B^+$ and $US^+$ should be subject to passive decay, presumably resulting in a slightly weakened association. Secondly, consistent with the literature on retrieval-induced forgetting[7–9], it is plausible to assume that actively retrieving the memory trace between the target stimulus $CS_A^+$ and $US^+$ would additionally weaken the memory trace between the competitor stimulus $CS_B^+$ and $US^+$. Both of the above mechanisms would lead to a differentiation of the memory engrams encoding $CS_A^+$ and $US^+$, and $CS_B^+$ and $US^+$ and should be reflected in diminished PRE to POST similarity. For the pair of control stimuli ($CS_A^- - US^0/CS_B^- - US^0$ and $CS_A^- - US^+/CS_B^- - US^+$), we did not expect changes in neural pattern similarity.

Preprocessing steps for multivariate fMRI analyses were identical as for previously mentioned univariate fMRI analyses, with the only exception that the functional imaging timeseries were not spatially smoothed. As for univariate fMRI analyses, we applied a conservative ICA to identify and remove obvious artefacts. GLMs were fitted into pre-whitened data space to account for local autocorrelations[50]. The individual level (first level) GLM design matrix per run and participant included 22 box-car regressors in total. Eighteen regressors coded for onsets and durations of all 18 presented CS–US-association trials (each modeled as single events of 1800 ms duration), one regressor coded for onsets and durations of the three within-run pauses (each 45 s), one regressor coded for onsets and durations of the attentional control search probes, two regressors coded onsets and durations of left and right button presses (delta stick functions on the recorded time of response button presses) and the six volume-to-volume motion parameters from motion correction during preprocessing were entered. Regressors were convolved with a hemodynamic response function ($\gamma$-function, mean lag = 6 s, SD = 3 s). Each first level GLM included one contrast to model activation related to each of the 18 presented CS–US-associations versus baseline (18 contrasts in total). A priori ROIs were built in MNI space and back-projected into subject native space using inverse normalization parameters obtained from FSL during preprocessing procedures. We used these individual ROIs for spatially constrained multivoxel pattern extraction from the respective contrast $t$-value maps. Similarity-based analyses were carried out using the MATLAB-based multivariate pattern analysis toolbox CoSMoMVPA[51]. We employed 1−Pearson's product–moment correlation coefficient ($1-r$) as a measure of pairwise dissimilarity between neural patterns of interest, separately for PRE and POST and the two ROIs. Within-subject pairwise neural dissimilarity was subtracted from 1 (to create a measure of neural pattern similarity) and Fisher-$Z$ transformed to closer approximate normally distributed data. We then calculated the within-subject PRE–POST change between the resulting pairwise neural pattern similarity measures (POST $r$ – PRE $r$, ∆Pearson's $r$). As we had a directional hypothesis of negative PRE–POST change of both $CS_A^0/CS_B^0$ and $CS_A^+/CS_B^+$ and to increase the number of trials included in the inference, we pooled neural pattern similarity measures across all patterns of interest. Lastly, average neural similarity changes were analyzed at the group level with one-sample $t$-tests against 0. Due to the expected negative effects of neural pattern similarity changes in patterns of interest, we used one-tailed tests accordingly. As there was no such directional hypothesis for the pairs of $CS_A^-/CS_B^-$, we employed two-tailed tests. We report effect sizes Cohen's $U3_1$ for one-sample $t$-tests against 0 (range: 0–1, 0.5 indicating no effect), calculated in the MATLAB-based Measures-of-Effect-Size-toolbox[41]. Based on the mean and

standard deviation of pattern similarity measures, we report post-hoc achieved power of all analyses.

**Univariate fMRI contrasts.**
$CS_A^0$ fMRI-RS contrast:

$$[2\times (CS_A^0 - US^0 - CS_B^0 - US^0)] - [(CS_A^0 - US^- - CS_B^0 - US^-) \\ + (CS_A^0 - US^+ - CS_B^0 - US^+)] \tag{1}$$

$CS_A^+$ fMRI-RS contrast:

$$[2\times (CS_A^+ - US^+ - CS_B^+ - US^+)] - [(CS_A^+ - US^- - CS_B^+ - US^-) \\ + (CS_A^+ - US^0 - CS_B^+ - US^0)] \tag{2}$$

Conjunction fMRI-RS contrast:

$$[2\times (CS_A^0 - US^0 - CS_B^0 - US^0)] - [(CS_A^0 - US^- - CS_B^0 - US^-) \\ + (CS_A^0 - US^+ - CS_B^0 - US^+)]$$

and

$$[2\times (CS_A^+ - US^+ - CS_B^+ - US^+)] - [(CS_A^+ - US^- - CS_B^+ - US^-) \\ + (CS_A^+ - US^0 - CS_B^+ - US^0)]. \tag{3}$$

**Computational models.** To formally characterize behavior in experiments 1, 2, 3, and 5, we fit six different variants of a reinforcement learning model using Rescorla–Wagner-like delta update rules[27]. For each experiment, we compared the six models that implemented different ways by which participants could have learned CS–US associative strength—and updated associative strength during choice-induced revaluation.

In model 1 (Pavlovian learning only), CS values are exclusively acquired during Pavlovian learning, without any further update during the choice-induced revaluation. On each trial, the value of the stimulus currently presented was updated according to the following rule:

$$V_{t+1,i} = V_{t,i} + \alpha_{\text{Learning}}(R_t - V_{t,i}) \tag{4}$$

where $\alpha_{\text{Learning}}$ is the learning rate, $V_{t,i}$ is the value of the $i$th stimulus (1–6 for the six CS), and $R_t$ is the reward value of the US (0, 0.5, and 1 for low-value, intermediate-value, and high-value outcome, respectively, and 0 if no outcome was presented) in the Pavlovian conditioning phase. CS values were initialized at 0.5. The estimated associative strength for each CS after the learning phase was directly passed to a softmax choice rule (Eq. (6)), without further modulation of CS associative strength by choice-induced revaluation.

Model 2 (Pavlovian learning and choice-induced revaluation, chosen CS) acquired stimulus values during Pavlovian learning exactly like model 1, but additionally updated CS–US associative strength by fictive reward prediction errors elicited by decisions in the choice-induced revaluation phase. The fictive reward prediction errors was based on our reasoning that presentation of CS during revaluation would lead to retrieval of the associated US and that, consistent with our hypothesis, stimulus–outcome association of the chosen CS would be strengthened, whereas the association of the unchosen CS would be weakened, resulting from nonmonotonic memory plasticity[16]. As no objective feedback (US) was presented during choice-induced revaluation, the reward prediction could only be derived by assuming associative retrieval. This model only updated associative strength of the chosen, but not the unchosen CS.

$$V_{t+1,\text{ch}} = V_{t,\text{ch}} + \alpha_{\text{ch}}(R_{t,\text{ch}} - V_{t,\text{ch}}) \tag{5}$$

where $\alpha_{\text{ch}}$ is the choice revaluation learning rate scaling the impact of fictive reward prediction errors elicited by fictive outcomes $R_{t,\text{ch}}$ (1 for chosen CS and −1 for unchosen CS) elicited by the US associated with each CS.

Model 3 (Pavlovian learning and choice-induced revaluation, chosen, and unchosen CS) was set up to account for the possibility that updating of both the chosen and unchosen CS association could occur during the choice-induced revaluation. It was identical to model 2, with the exception that it performed an update to the associative strength of the unchosen stimulus, using a separate learning rate $\alpha_{\text{unch}}$, in addition to updating the chosen CS associative strength.

These same three models were set up as variants (associative value models) that were identical in all respects except for the outcomes during Pavlovian learning, which were modeled as 0 (no outcome presented) or 1 (outcome presented) and CS associative strength values were scaled with the normalized (0–1), individually rated subjective value of the outcome (pre-rating) at the end of the Pavlovian learning phase.

The estimated associative strengths for all CS after the choice-induced revaluation phase were passed to a softmax decision function to generate choice probabilities for each option on each trial:

$$P_{C,t} = \frac{1}{1 + \exp(-\text{VD}_t/\tau)} \tag{6}$$

where $P_{C,t}$ is the model's probability to select option $C$ on trial $t$, the choice the participant actually made on trial $t$. $\text{VD}_t$ is the value difference (or difference in associative strength) between the chosen and unchosen CS on trial $t$, and $\tau$ is a temperature parameter that determines the degree of stochasticity in participants' choice behavior.

To find the free parameters that best described participants' behavior, we used a two-step fitting procedure to minimize the negative log-likelihood estimate (−LLE):

$$-\text{LLE} = -\Sigma \log(P_{C,t}) \qquad (7)$$

First, we ran a grid search on an $n$-dimensional grid in log space (where $n =$ number of free parameters for each model), with 30 steps along each dimension. The grid optimum was then used as initial value and passed to constrained non-linear optimization using the MATLAB function fmincon. Optimized negative log likelihoods were compared by means of the sample-size corrected Akaike Information Criterion (AICc, Eqs. (8) and (9)). The model with the lowest AICc value was considered best for the observed participants' choice data, penalized for model complexity and sample size (number of participants).

$$\text{AIC} = 2k - 2(-\text{LLE}) \qquad (8)$$

where $k$ is the number of free parameters of the model and −LLE is the negative log likelihood of the model given the data.

$$\text{AIC}_C = \text{AIC} + \frac{2k^2 + 2k}{n - k - 1} \qquad (9)$$

where $k$ is the number of free parameters in the model, and $n$ is the sample size.

Additionally, we performed 10,000 simulations of choice behavior per participant. After value estimation as described above, using individual parameters of the best-fitting models at group level, the resulting value estimates for each stimulus were entered in the exact same sequence of 120 choices that the participants individually experienced during the decision probe phase. Simulated overall choice probabilities were averaged per participant across 10,000 simulations.

**Reporting summary**. Further information on research design is available in the Nature Research Reporting Summary linked to this article.

## Data availability

The raw behavioral data, univariate parameter estimates extracted from regions of interest, thresholded and unthresholded univariate $Z$-maps, neural pattern similarity correlation matrices and brain–behavioral correlation data that support the findings of this study are publicly available at GitHub (https://github.com/LLuettgau/revaluation). The neuroimaging raw data that support the findings of this study are available upon reasonable request from the corresponding author (L.L.). The neuroimaging raw data are not publicly available due to them containing information that could compromise research participant privacy/consent. The source data underlying Figs. 1e–i, 2d, e, 3a, b, 4b and Supplementary Figs. 1a–h, 2a–h, 3a, b, and 5a are provided as a Source Data file. A reporting summary for this article is available as a Supplementary Information file. Source data are provided with this paper.

## Code availability

Custom analysis code for the reported behavioral data analyses, univariate fMRI analyses on extracted parameters, multivariate fMRI analyses on neural pattern similarity correlation matrices, brain–behavioral correlational data analyses and computational modeling/simulations are publicly available at GitHub (https://github.com/LLuettgau/revaluation). Source data are provided with this paper.

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

## Acknowledgements

The authors thank all volunteers who participated in this study. The authors would like to thank Halla Mulla-Osman, Stefanie Linnhoff, Nicola Harzen and Denise Scheermann for their invaluable support during data acquisition. We thank the Center for Magnetic Resonance Research of the University of Minnesota for providing the Multiband accelerated fMRI sequence. The authors additionally thank Theo O.J. Gruendler, Lukas M. Neugebauer, and Daniel Priegnitz for helpful discussions during set-up of the study and for sharing custom MATLAB code. This work was funded by the federal state of Saxony-Anhalt and the "European Regional Development Fund" (ERDF 2014-2020), Vorhaben: Center for Behavioral Brain Sciences (CBBS), FKZ: ZS/2016/04/78113.

## Author contributions

L.L. designed the study and conceptualized research, acquired the data, analyzed the data, drafted the manuscript, read and edited versions of the manuscript and approved the final version of the manuscript. C.T. set up the MRI acquisition protocol, read and edited versions of the manuscript and approved the final version of the manuscript. L.F.K. analyzed the data, read and edited versions of the manuscript and approved the final version of the manuscript. G.J. designed the study and conceptualized research, analyzed the data, read and edited versions of the manuscript and approved the final version of the manuscript.

## Competing interests

The authors declare no competing interests.
