## [Peer Review File · Nature Communications]

Reviewers' Comments:

Reviewer #1:

Remarks to the Author:

Luettgau and colleagues report the results of a set of experiments examining the mechanisms of choice-induced preference changes. This manuscript includes three behavioral experiments and one behavior + fMRI experiment. Together the data suggest (to me) that choice-induced changes in preferences over conditioned stimuli result, in part, from changes in CS representations in the hippocampus. There are a number of clever design features in this work and the results are very interesting and thought provoking. The role of hippocampal representations of stimuli with regard to choice induced preference changes is novel and intriguing. Overall, I think this could be a nice paper with some relatively straightforward modifications to the text/claims. However, although easily made these changes to the paper are very important in my view. While I think that there are novel and important conclusions that can be drawn from this work, I don't find the authors' current claims satisfactorily supported by the data.

1. I am guessing that the authors really mean to claim that choice-induced changes in preferences over conditioned stimuli result, **in part**, from changes in the strength of CS-US pairings in the hippocampus. However, the current manuscript text seems to make the much stronger – and unjustified – claim that changes in the associative strength of CS-US pairs explain ALL of the choice-related effects. Only experiment 4 tests the strength of CS-US pairs before and after the critical choice manipulation. The repetition suppression (RS) tests show that there is a significant change in one of the two relevant conditions. Moreover, they find a significant **correlation** between the size of RS effects and choice-induced changes in preferences. For the moment, let's assume that the RS effects are specific to changes in association strength (I'll argue below that this is not necessarily true). Even if we take this as given, this correlation is rather modest (spearman rho = 0.31), indicating that much of the individual variability in behavior is not explained by changes in associative strength as indexed by differences in RS. This modest correlation is not at all surprising given the noise inherent in both fMRI and choices, as well as the non-specific anatomical mask used to extract the fMRI data. Thus, the links between hippocampus activity and choice-induced preference changes are an interesting and, in my opinion, important result. Still, it indicates that it is most likely that changes in associative strength cannot fully explain the choice effects on subsequent revealed preferences. It seems that additional factors are likely to be in play as well, but the current text does not explicitly acknowledge or discuss this possibility. Thus, I strongly suggest that the authors rephrase their claims throughout the manuscript and conclusions in the discussion to state that changes in associative strength partially explain changes in choice-induced preferences.

2. Related to point 1 above, the analysis attempting to rule out changes in the cached values of the CS is not at all convincing to me. On the contrary, it suggests to me that RS effect in the hippocampus can be based on either the values and/or identities of the two stimuli, in other words it is not specific to associative strength. The authors basically attempt to use the RS effects to show that the size of suppression between the CS0 A that wasn't chosen and the

US isn't related to choice induced-preference changes. The idea being that the US- represents a low value and if unchosen CS0 A also has a lower value, then it will then there will be suppression between it and the US-. Thus, this is essentially a test of the idea that the hippocampus (or any other brain region) is storing an abstract representation of general value (regardless of a specific memory or association). This is in contrast to idea behind using RS as a test of associative strength, which is that because of the Pavlovian stage, an association is formed and then when the CS is presented it elicits activity related to the associated US (as they put it "conditioning enhances CS's ability to pre-activate neural ensembles coding for US"). Naturally, RS may reflect *both* suppression due to identity and value similarity – which is an important possibility that the authors never test or discuss. However, given the results as a whole, I think they should discuss it.

a. Setting aside this ignored third possibility for now, there is a critical flaw in using CS0 A for the value-suppression test. In the fMRI sample, there was no difference in the probability of choosing CS0 A vs CS0 B after CS0 A was supposed to be devalued/disassociated in the choice phase, i.e. no choice-induced preference change. Thus, the fact that there is no significant change in CS0 A – US- suppression in PRE vs POST is irrelevant.

b. Furthermore, the positive correlation between the "Cached value control analysis fMRI-RS contrast" and preferences for CS0 A makes sense when considering that this contrast is an approximate opposite of the CS0 A fMRI-RS contrast. Remember that the CS0 A fMRI-RS contrast showed a negative correlation with preferences for CS0 A. Beyond the sign of the correlation, the fact is that these two complex contrasts contain the same regression coefficients, but with different signs and weights (see below). Ultimately, I don't see how we can say with any certainty how exactly the six regression coefficients in each contrast are contributing to the observed results. Both contrasts contain components that one could argue measure changes in association identity as well as components that would measure changes in value. The authors make these very arguments themselves in using the two contrasts the way they do in the current text. Thus, I think the key result in this work is that choosing for or against a stimulus changes the way it is represented in the hippocampus. Whether this hippocampal activity is related to CS-US associations, more general valuation, or something else is not established by the current analyses.

CS0 A fMRI-RS contrast:

$$[2 \times (CS^0_A-US^0 - CS^0_B-US^0)] - [(CS^0_A-US^- - CS^0_B-US^-) + (CS^0_A-US^+ - CS^0_B-US^+)] \quad (1)$$

"Cached" value control analysis fMRI-RS contrast:

$$[2 \times (CS^0_B-US^- - CS^0_A-US^-)] - [(CS^0_A-US^0 - CS^0_B-US^0) - (CS^0_B-US^+ - CS^0_A-US^+)] \quad (4)$$

3. On a different note, more discussion of the theoretical background would be beneficial. The authors mention two different memory theories: reconsolidation and suppression via retrieval induced forgetting. It would help the general audience at Nat Comms if they expand on what they think is happening to lead to both sides of the phenomenon (strengthening and weakening of the traces). Do they think these are both operating via joint mechanisms (as is the premise with their conjunction approach in the imaging) or different ones (more in line with separating these two theories as is implied in the text)? Is reconsolidation here supposed to work mostly to strengthen the trace (the weakening of memory traces during reconsolidation is more generally tested/assumed so references reporting that reconsolidation strengthens the memory independent from repetition would be useful) and retrieval induced forgetting to weaken the trace? Some elaboration on this theoretical part would be very helpful.

4. The authors should also conduct and report tests of whether or not the models RW3 and assoc 3 (i.e. the best-fitting models) can produce patterns of behaviour similar to that observed in real participants when using the best-fitting model parameters to generate simulated data. This will allow us to see if the model is a reasonable candidate for generating the behaviour in addition to being the best amongst the chosen candidate models.

5. The correlation between hippocampal RS effects and choice-induced preference shifts is referred to as the hippocampus predicting the choice effects. Unfortunately, the term prediction is used in a variety of ways, but I take issue with referring to this correlation as a prediction. I think it is best to reserve the term prediction for out-of-sample tests, which the current analysis is not. It would be rather simple to compute an out-of-sample test of the predictive accuracy of hippocampal signals with regard to preference shifts if the authors wish to back up the claims about prediction. Otherwise, I request that they stick to calling this result a correlation.

More minor points:

1. Some of the tests are one sided, while others are two-sided. This seems justified, but these distinctions and justifications are somewhat buried in the methods section where many readers will miss them. I think it would be useful to mark the 1 and 2-sided tests where they appear in the results section.

2. Is the following an error? That seems likely given the exact same values – although it may be that they are indeed the same. “we found decreased associative strength for CS0A ($Z = 2.27$, $P = 0.023$, Wilcoxon signed-rank test) and increased associative strength for CS+A ($Z = 2.27$, $P = 0.023$, Wilcoxon signedrank test) during POST”

3. What exactly is the difference between Fig 3B and S2A? They both seem to have the same contrast (they even specify eq. 3) but different values. Please clarify.

4. This sentence refers to the wrong equation, “We investigated post choice induced reevaluation conjunction effects (Equation 9)”

Reviewer #2:

Remarks to the Author:

This study explore choice-induced revaluation of previously learned associative values in 3 behavioral studies and an fMRI study that uses repetition suppression. The overall finding consistent across all experiments is that the associative value is strengthened, if the CS has been previously chosen and was associated with a higher-valued US, and is decreased if it has not been selected in the recent past and it was associated with a lower-valued US. This later effect appears to be not as strong as the former. Consistent effects are found in the hippocampus and the lateral OFC, which are also positively correlated with future choices.

This is a well-designed study that investigates an important research question with implications for real-life decision-making problems that humans face every day. The design is intricate and sometimes a bit hard to follow, but it follows a rational logical tailored to the phenomenon of repetition suppression. I think the research question is broad enough to be interesting and relevant for the diverse readership of Nature Communications even outside the field of Cognitive Neuroscience. The analyses are planned with great care to the details and the rational approach provides a good thread to follow the logic of the design. I don't have any principle objections to this paper, but there are a few issues that I would like to see addressed.

First, some of the behavioral (but also the neuroimaging) effects appear to be not as strong given that they become significant with one-tailed testing. While there is nothing wrong with this in principle, I think the use of one-tailed test needs better preparation and justification in the description of the results and also in the Methods section, i.e. the directionality of the hypothesis needs to be stated more clearly and then should be followed through in all contrasts. Right now, the selection of one-tailed or two-tailed tests appears to be mainly driven by the size of the effect (if strong, then two-tailed test are reported, if weak, then the authors use one-tailed tests). This needs to be stratified. Another consideration that always comes up when using one-tailed tests is that of statistical power. It would be good, if the authors could provide a measure of effect size in the main text along with a measure of the (post-hoc) statistical power, so that the reader can get an impression of the strength of the effect was influenced by the power in the experiment for both the behavioral and neuroimaging data since one-tailed tests are also prevalent in the ROI analyses of the brain data.

Second, the fMRI experiment different from the preceding behavioral experiment in the introduction of the RS runs PRE and POST choice revaluation. In these runs, no outcomes are presented following the respective CS presentations. This opens the possibility for extinction processes to kick in during these runs that could affect the changes induced by choice revaluation. Such possible confounds should be critically examined in the Discussion.

Third, this experiment connects to Tom Schonberg's work on preference change through mere choices in a significant way, although this study is more multi-dimensional in its design and implications. Nevertheless, it would be good to connect to this literature and highlight the similarities and differences between Schonberg's approach and the present study.

Minor points:

In the description of the fMRI findings (line 171 onwards), the contrast estimates are named "associative strength". This seems to be a bit suggestive of a cognitive process (i.e. you cannot directly measure associative strength through contrasting two BOLD activations). A straight description of the interaction would be more appropriate, which then be interpreted as associative strength.

line 567: "participant was included"-> remove "was"

line 586: please spell out the meaning of COPE

Reviewer #3:

Remarks to the Author:

The manuscript describes a set of findings from 4 experiments (3 behavioural and 1 fMRI) investigating how the current act of making a choice will bias future choices. The authors specifically test the hypothesis that this decision-induced bias is due to changes in the associative memory structure underlying participants' choices. The basic paradigm consists of 3 phases: during Pavlovian conditioning, 6 originally neutral stimuli (CS) are associated with outcomes such that 2 CS each are linked to a high (+), medium (0) or low (-) value outcomes. During the revaluation phase, participants repeatedly make choices between two critical, pre-defined CS (e.g. in Exp. 1 and 4, they chose between one high-value CS+ and one medium-value CS0). In a final decision probe phase, participants then make choices between all possible pairings of CS. Behaviourally, on this final probe test they are now more likely to choose the CS they had selected repeatedly during the preceding revaluation phase, compared to the other same-value CS that never appeared during revaluation. Similarly, they are less likely to now choose the CS they had repeatedly "un-selected" during revaluation, compared to the other same-value CS. The fMRI data suggest that this change in preference is due to a strengthening and weakening of the respective stimulus-outcome associations.

The paper is well written, coherently argued to the largest degree (see comments below though), and certainly an interesting read. Clear predictions are made, and tested with appropriate methods. As a memory (and not decision making) expert, I am unable to judge the novelty of the findings within the decision-making field. From the introduction, I understand that decision-induced biases have been shown before, and the novel aspect here is the demonstration that changes in associative strength underlie such biases. The conceptual advance from this paper thus critically depends on the degree to which the authors can show conclusive evidence for such a memory-based mechanism.

Major questions and comments:

(1) To analyse the choice biases behaviourally, the authors report that "For the decision probe phase, we computed an average overall CP per conditioned stimulus per subject including all binary decisions in which the respective CS was present" (p.20/21). I would like to see this behavioural data unpacked a little. In particular, I am assuming that the major effect will come from probe trials where subjects have to make a choice between two same-value CS, because otherwise they will choose the higher-valued CS on the vast majority of trials? At least it would be surprising if the strengthening/weakening was pronounced enough to override the conditioned value (e.g. when subjects select between CS with different associated values, e.g. CS+ and CS0). Relatedly, is the choice bias mainly produced by the 8 repetitions where participants directly choose (again) between the two CS that were the critical stimuli in the previous revaluation phase? If not, what types of trials does it mainly generalize to?

(2) Regarding the fMRI data, the authors used a set of tightly controlled but quite complex contrasts, designed to isolate the specific effects of prior choice on associative strength, as measured by repetition suppression. It would be reassuring for the reader to see that within these complex contrasts, the effects in the hippocampus and OFC are driven by the expected components. That is, show (maybe in a supplementary figure) the degree of repetition suppression when the critical CS (e.g. CSA+ for strengthening) is paired with its original associate, compared to when it is shown with any of the other associates, separately for strengthening and weakening. If I understand Fig. 2 correctly, it depicts the difference between trials where the CSA was paired with its original outcome,

and trials where the CSB was paired with its original outcome. Unpacking the pattern underlying these differences would help to determine whether there are any basic effects other than repetition suppression, e.g. exposure frequency, prediction error when CS are followed by the wrong outcome etc., that could influence the hippocampal signal. At current, I find it difficult to extract all these alternative possibilities from the data presented, and to see the direct evidence for mnemonic reactivation of the US.

(3) Related to the previous point, the authors make the strong claim that changes in associative strength underlie the observed effects. Given this aspect seems to be critical in terms of novelty, it would be reassuring to see conceptually similar results replicated with a complimentary analysis, e.g. a multivariate analysis that directly decodes the (presumably mentally reinstated) outcome during each CS presentation. Of course, on correct trials of the fMRI-RS task, the outcome will be visually presented and thus strongly present in brain activity. there are plenty of repetitions in this task, however, where incorrect outcomes are shown, and where we would still expect participants to associatively retrieve the correct outcome, given they have to make correct-incorrect judgments during this phase. This could be shown e.g. using a representational similarity approach comparing the neural patterns elicited during CSA and CSB trials (minus the ones that were followed by the original outcome), or using pattern classifiers with the same underlying logic.

(4) In some instances, I found it difficult to follow the exact reasoning regarding the hypothesized memory mechanism underlying the observed effects, and the paper could be improved if a more explicit and coherent explanation of this mechanism was offered. Based on the memory literature, I initially found it surprising, in the context of this study design, that the authors argue for a weakening of the unselected CS-US association, since the respective CS are still presented multiple times during the revaluation phase, presumably encouraging participants to repeatedly recall the associated values of both choice stimuli (e.g., CS+A and CS0A in Exp. 1 and 4) in order to make a value-based choice between them. In theory, repeated recall of both associations should lead to strengthening of both associations. Is it possible that participants simply learn a "do-not-choose-this-stimulus" tag along with the unselected CS during revaluation, rather than weakening the CS-US association of the unselected stimulus? I understand that the fMRI data do not support such a tagging explanation, but seeing additional (e.g. behavioural) evidence against it would be reassuring. Second, given the study design, it seems equally likely that repeatedly selecting CSA+ would weaken the CSB+ association, since the CSB+ is associatively linked to the same outcome and can thus be considered a competitor, maybe more so than the CSA0. Again, a more explicit argument regarding the specific memory mechanism, potentially supported by empirical evidence, would strengthen the conclusions that can be drawn from this study.

(5) In the fMRI-RS task, participants were asked to make memory judgments on whether the CS-US combination matched the originally learned pairing. It would be interesting to know if the behavioural data during this phase provide any evidence for strengthening/weakening of the critical, revalued associations, either in terms of percent correct (which might be at ceiling, with the reported mean of 92%), or in terms of reaction times.

(6) The behavioural data from the fMRI experiment do not replicate the behavioural effect found in Exp. 1 and 2, in particular the weakening of the unselected CS. The authors suggest that this null effect might be due to reconsolidation via re-exposure during the fMRI-RS task. It is difficult to perceive how reconsolidation could play a role at these short, within-session time scales, given that reconsolidation in the strict sense can only occur for memories that were consolidated (e.g. by 48h delay, sleep etc.) in the first place. The authors should at minimum elaborate on this explanation. A more parsimonious, alternative view is that, consistent with the memory literature (e.g. Hulbert & Norman, CerebCortex; Storm, Bjork & Bjork, 2008), is that retrieval-induced weakening can be

eliminated or even reversed into strengthening when interleaved with, or followed by, re-exposure to the weakened representations.

Minor comment:

- Figure 2 would benefit from a visualization of which comparisons (interaction, paired tests) were significant in the bar graphs shown at the bottom.

We would like to thank the reviewers for their detailed and thoughtful comments. They have led to a substantially revised manuscript. The reviewers' comments inspired a set of changes, including changes to the manuscript, conduction of further analyses and an entirely new behavioral experiment. We believe that the changes we made in response to the reviewers' comments and suggestions represent a substantial improvement over our initial submission. We are therefore grateful for the time and effort the reviewers put into our paper.

We reproduce each of the reviewers' comments below in italics before our response to each point. Changes made to the manuscript are in blue, both here and in the manuscript.

Reviewer #1

“There are a number of clever design features in this work and the results are very interesting and thought provoking. The role of hippocampal representations of stimuli with regard to choice induced preference changes is novel and intriguing.”

We thank the reviewer for this overall positive assessment of our manuscript, but also for suggesting additional simulation-based analyses and carefully spotting some points that required re-phrasing of the claims and critical discussion of employed control analyses.

1) *“I am guessing that the authors really mean to claim that choice-induced changes in preferences over conditioned stimuli result, *in part*, from changes in the strength of CS-US pairings in the hippocampus. However, the current manuscript text seems to make the much stronger – and unjustified – claim that changes in the associative strength of CS-US pairs explain ALL of the choice-related effects. Only experiment 4 tests the strength of CS-US pairs before and after the critical choice manipulation. The repetition suppression (RS) tests show that there is a significant change in one of the two relevant conditions. Moreover, they find a significant *correlation* between the size of RS effects and choice-induced changes in preferences. For the moment, let's assume that the RS effects are specific to changes in association strength (I'll argue below that this is not necessarily true). Even if we take this as given, this correlation is rather modest (spearman rho = 0.31), indicating that much of the individual variability in behavior is not explained by changes in associative strength as indexed by differences in RS. This modest correlation is not at all surprising given the noise inherent in both fMRI and choices, as well as the non-specific anatomical mask used to extract the fMRI data. Thus, the links between hippocampus activity and choice-induced preference changes are an interesting and, in my opinion, important result. Still, it indicates that it is most likely that changes in associative strength cannot fully explain the choice effects on subsequent revealed preferences. It seems that additional factors are likely to be in play as well, but the current text*

does not explicitly acknowledge or discuss this possibility. Thus, I strongly suggest that the authors rephrase their claims throughout the manuscript and conclusions in the discussion to state that changes in associative strength partially explain changes in choice-induced preferences.”

We completely agree with the reviewer that from the strength of the correlation alone, it is clear that we cannot explain all of the variance in participants' behavior with the observed changes of CS-US associative strength in the hippocampus. To emphasize this and to prevent an overstatement of our findings, we have therefore reworded the paragraphs in question by emphasizing that changes in associative strength partially explain choice-induced preference changes, thus toning down our claims throughout the manuscript.

*2) “Related to point 1 above, the analysis attempting to rule out changes in the cached values of the CS is not at all convincing to me. On the contrary, it suggests to me that RS effect in the hippocampus can be based on either the values and/or identities of the two stimuli, in other words it is not specific to associative strength. The authors basically attempt to use the RS effects to show that the size of suppression between the CS0A that wasn’t chosen and the US isn’t related to choice induced-preference changes. The idea being that the US- represents a low value and if unchosen CS0A also has a lower value, then it will then there will be suppression between it and the US-. Thus, this is essentially a test of the idea that the hippocampus (or any other brain region) is storing an abstract representation of general value (regardless of a specific memory or association). This is in contrast to idea behind using RS as a test of associative strength, which is that because of the Pavlovian stage, an association is formed and then when the CS is presented it elicits activity related to the associated US (as they put it “conditioning enhances CS’s ability to pre-activate neural ensembles coding for US”). Naturally, RS may reflect *both* suppression due to identity and value similarity – which is an important possibility that the authors never test or discuss. However, given the results as a whole, I think they should discuss it.*

a. Setting aside this ignored third possibility for now, there is a critical flaw in using CS0A for the value-suppression test. In the fMRI sample, there was no difference in the probability of choosing CS0A vs CS0B after CS0A was supposed to be devalued/disassociated in the choice phase, i.e. no choice-induced preference change. Thus, the fact that there is no significant change in CS0A – US- suppression in PRE vs POST is irrelevant.

We thank the reviewer for the thoughtful concern that the repetition suppression signal (RS) could indicate diminished BOLD responses due to value or identity repetition or associative strength, which is consistent with empirical results (e.g. Klein-Flugge, Barron, Brodersen, Dolan, & Behrens, 2013). We never meant to claim any exclusiveness for the proposed associative interpretation to account for all facets of the RS signal and have thus now amended alternative explanations for potential underpinnings of RS signals in the introduction of the “cached value” analysis (p. 16), for instance in this paragraph that we reproduce below:

By its very nature, the fMRI-RS signal for learned CS-US associations represents BOLD signal reductions due to repetition of both value and identity features shared by CS and US, alongside the associative strength between CS and US (Klein-Flugge, Barron, Brodersen, Dolan, & Behrens, 2013). However, CS^0_A should by design of the experiment not be capable to elicit any associative strength- or identity-related RS effects when followed by US^- , as it was never coupled with US^- during Pavlovian conditioning. Since value and identity of the US are inextricably linked in the present design, we reasoned that any RS signal changes for CS^0_A followed by US^- from PRE to POST would most likely be attributable to changes in valuation of CS^0_A .

However, we do not believe that US value and identity are mutually exclusive in our design – they are in fact inextricably linked – and we therefore do not think the reviewer’s concern necessarily contradicts the proposed associative mechanism. To the contrary: both value or identity are learned associatively (at least in the present study) and presumably have to be retrieved in an associative fashion. Our reasoning in applying the two contrasts in the way we did was that both CS^0_A and CS^0_B should (by design of the experiment) not be capable to elicit a US identity- or association-dependent RS effect when followed by US^- , as they were never followed by US^- during Pavlovian learning. However, the expected reduction of choice probability of CS^0_A (if present) could alternatively have resulted from reduction of “cached value” of CS^0_A by non-choices of this stimulus during revaluation. Therefore, we expected that CS^0_A representations would become more similar to US^- , a low value outcome (“general value” as the reviewer suggests). However, we agree with the reviewer that our reasoning had not been coherent in terms of the “cached value” contrast and the implied assumptions. The “cached value” analysis would have critically depended on decreased choice probability of CS^0_A , which was not present in the fMRI experiment. We nevertheless believe that it is hard to conceive how this specific RS effect could be explained by associative retrieval or identity of US^- , since CS^0_A should by design not be able to pre-activate a representation (identity features) of US^- . We hope that the changes we applied in response to your comment make this clearer to the reader.

b. Furthermore, the positive correlation between the “Cached value control analysis fMRI-RS contrast” and preferences for CS_{0A} makes sense when considering that this contrast is an approximate opposite of the CS_{0A} fMRI-RS contrast. Remember that the CS_{0A} fMRI-RS contrast showed a negative correlation with preferences for CS_{0A}. Beyond the sign of the correlation, the fact is that these two complex contrasts contain the same regression coefficients, but with different signs and weights (see below). Ultimately, I don’t see how we can say with any certainty how exactly the six regression coefficients in each contrast are contributing to the observed results. Both contrasts contain components that one could argue measure changes in association identity as well as components that would measure changes in value. The authors make these very arguments themselves in using the two contrasts the way they do in the current text. Thus, I think the key result in this work is that choosing for or against a stimulus changes the way it is represented in the hippocampus. Whether this hippocampal activity is related to CS-US associations, more general valuation, or something else is not established by the current analyses.”

We agree with the reviewer that the “cached value” contrast indeed somewhat appears like an approximate opposite of the “associative” CS_{0A}-CS_{0B} contrast (Equation 1). However, we would like to point out that, contrary to what might be expected, the parameter estimates for the associative (Equation 1) and the cached value contrast (Equation 4) are not negatively correlated. In the left hippocampus, if anything, there is a positive correlation, which however is not significant (PRE: $\rho = .22$, $P = .170$; POST: $\rho = -.03$, $P = .852$; Spearman correlations, two-tailed). Therefore, we would think it is unlikely that this contrast merely shows the inverse of the associative contrast. We also like to kindly note that, unlike stated by the reviewer, we had not reported a negative correlation of the CS_{0A} fMRI-RS contrast with preferences for CS_{0A} in our initial submission. Indeed, there is no such significant negative correlation between the POST hippocampal CS_{0A} associative contrast (Equation 1) parameter estimates and CS_{0A} choice bias (overall CP CS_{0A}: $\rho = .07$, $P = .663$; difference of overall CP CS_{0A} and CS_{0B}: $\rho = -.10$, $P = .550$; Spearman correlations, two-tailed). Therefore, to us, the reported positive correlation between the “cached value” effect in the right hippocampus and the choice probability of CS_{0A} was rather unexpected.

Regarding the reviewer's point on whether the change in the hippocampal representation is due to alterations in CS-US association, representation of the stimulus per se, or something else is, in our view, likely hard to tease apart experimentally. This is because in a learning context, any changes, even if linked purely to stimulus presentation, inevitably might also be due to the stimulus evoking linked representations. However, we would argue that it is hard to conceive that a mere change in stimulus representation would (i) only be expressed when presenting the "correct" (learnt) CS-US associations, and, perhaps more

importantly (ii) reverse direction depending on choice during revaluation (decrease for CS⁰-US⁰, increase for CS⁺-US⁺), as observed in the data. We nevertheless agree with the reviewer that the hippocampal RS effects could be related to associative retrieval of US value or identity features. We have added a paragraph to the discussion addressing this issue that we reproduce below for the reviewer's convenience, but we would also like to point out that this would be an important research question in its own right - which however goes beyond the scope of the present study.

Even though our data provide evidence for choice-induced changes to associative strength, the current approach does not allow to dissociate which exact features of the US contribute to repetition suppression effects. As US value and identity are inextricably linked in our experiment, the observed effects could be related to the quantitative or qualitative changes in CS-dependent pre-activation of identity- or value-related features of the US. However, both identity- and value-related features of the US are learned associatively in the present study and are therefore quite likely to be retrieved in an associative fashion to guide choices. Alternatively, the observed change in the hippocampal repetition suppression could be due to choice-induced alterations of CS representations per se. However, changes of CS representation cannot explain the specificity of RS signals to presentations of the "correct" (learnt) CS-US associations, and, perhaps more importantly, cannot account for the reversed directionality of RS effects depending on choice during revaluation (decrease for CS⁰-US⁰, increase for CS⁺-US⁺), as observed in the data.

3) *“On a different note, more discussion of the theoretical background would be beneficial. The authors mention two different memory theories: reconsolidation and suppression via retrieval induced forgetting. It would help the general audience at Nat Comms if they expand on what they think is happening to lead to both sides of the phenomenon (strengthening and weakening of the traces). Do they think these are both operating via joint mechanisms (as is the premise with their conjunction approach in the imaging) or different ones (more in line with separating these two theories as is implied in the text)? Is reconsolidation here supposed to work mostly to strengthen the trace (the weakening of memory traces during reconsolidation is more generally tested/assumed so references reporting that reconsolidation strengthens the memory independent from repetition would be useful) and retrieval induced forgetting to weaken the trace? Some elaboration on this theoretical part would be very helpful.”*

During revision of the manuscript, we realized that our theoretical reasoning regarding reconsolidation as a mechanism underlying choice-induced preference changes had not been

coherent and actually not well in accord with the literature on reconsolidation. We have thus removed this line of reasoning from the manuscript. We have updated the introduction with a more parsimonious explanation for both choice-induced strengthening and weakening of the associative traces. In naturalistic decision making scenarios, choices often have to be made without direct experience of feedback or rewards. Instead, decision makers have to rely on relational knowledge of stimuli, actions and outcomes. The absence of direct external feedback and the resulting inability to adjust synaptic weights based on error-driven learning mechanisms suggests the use of unsupervised learning to optimize behavior in those situations. Likely candidate mechanisms for such unsupervised behavioral adaptation are memory retrieval dynamics. It is well established that retrieval of an item from memory, e.g. a conditioned stimulus (CS) triggering retrieval of an associated outcome, leads to improved remembering. However, memory for competing items, e.g. a CS associated with the same outcome, is impaired simultaneously (Anderson, Bjork, & Bjork, 1994; Hulbert & Norman, 2015; Wimber, Alink, Charest, Kriegeskorte, & Anderson, 2015). Such retrieval-induced forgetting (Anderson et al., 1994) would predict choice biases towards previously chosen CS based on retrieval-related strengthening of CS-US association. However, retrieval-induced forgetting would predict the same effect for a previously presented, but unchosen CS, since both chosen and unchosen CS activate neural populations representing the respective associated outcome (Barron, Dolan, & Behrens, 2013; Boorman, Rajendran, O'Reilly, & Behrens, 2016; Howard, Kahnt, & Gottfried, 2016; Klein-Flügge et al., 2013; Onat & Büchel, 2015; Tonegawa, Morrissey, & Kitamura, 2018). A more recent theoretical framework (Nonmonotonic Plasticity Hypothesis, as reviewed in Ritvo, Turk-Browne, & Norman, 2019) suggests U-shaped spreading activation during associative memory retrieval: Inactive memories remain unaltered, whereas moderately activated associative memories are weakened, and higher activation leads to strengthening of memories (Ritvo et al., 2019). Translating this idea to memory-based decisions between two CS, we assumed that both CS would moderately activate neural populations representing the associated outcome (as the outcome is never presented). However, consistent with the finding that chosen options receive higher attentional weighting than unchosen options (as reflected in higher learning rates for chosen options (Klein, Ullsperger, & Jocham, 2017; Palminteri, Khamassi, Joffily, & Coricelli, 2015)), we further assumed that choices of a CS would induce additional activation of the associated outcome, whereas this would not be the case for unchosen CS, retaining an intermediate activation state of the associated outcome. Thus, we hypothesized that choosing a CS would strengthen the related stimulus-outcome association. Contrarily, not choosing a CS would weaken the respective stimulus-outcome association. We expected that these choice-induced alterations of the associative memory structure would result in subsequent preference changes. In other words, we assumed that choices themselves can act as self-generated “teaching signals”,

dynamically altering stimulus-outcome associations stored in memory by shifting associative memories along a nonmonotonic plasticity function (Ritvo et al., 2019).

We have modified the corresponding paragraphs in both the introduction and discussion of the manuscript. We thank the reviewer for making us address this more clearly.

4) *“The authors should also conduct and report tests of whether or not the models RW3 and assoc3 (i.e. the best-fitting models) can produce patterns of behaviour similar to that observed in real participants when using the best-fitting model parameters to generate simulated data. This will allow us to see if the model is a reasonable candidate for generating the behaviour in addition to being the best amongst the chosen candidate models.”*

Thank you for pointing this out. Following your suggestion, we have simulated (using the best-fitting parameters of the respective models) the 120 probe phase choices per participant 10,000 times each and then averaged the resulting 10,000 simulations per subject. We have added these results as new Supplementary Figure S1 (E-H). In brief, it shows that the simulated choices qualitatively recapitulate most of the observed empirical choice pattern, (with the exception of the observed reduced choice probability of CS_B^0 in Experiment 4, (Supplementary Figure S1D). We conclude that our models successfully reproduce the pattern of behavior to which they had been fitted. We insert supplementary figure S1 here for convenience:

Figure 1. A-D) Empirical choice results (as in manuscript Fig. 1E-H). Previously chosen CS (blue scatter) are selected more often compared to equivalent CS (black scatter) in Experiments 1, 2, 4 (E, F, H) and previously non-chosen CS (yellow scatter) are selected less often compared to equivalent CS (black scatter) in Experiment 1, 2 (E, F) during decision probe. The effect is not present in Experiment 3 (G), indicating that the roughly equal proportion of choices and non-choices of CS_A^0 during revaluation had cancelled each other out. E-H)

Averaged simulated choice probabilities (10,000 simulations per participant), recapitulating observed empirical choice patterns.

5) *“The correlation between hippocampal RS effects and choice-induced preference shifts is referred to as the hippocampus predicting the choice effects. Unfortunately, the term prediction is used in a variety of ways, but I take issue with referring to this correlation as a prediction. I think it is best to reserve the term prediction for out-of-sample tests, which the current analysis is not. It would be rather simple to compute an out-of-sample test of the predictive accuracy of hippocampal signals with regard to preference shifts if the authors wish to back up the claims about prediction. Otherwise, I request that they stick to calling this result a correlation.”*

The reviewer is of course perfectly right here. The reported correlation is not an out-of-sample test. Our intention was not to make any claims about out-of-sample prediction of the result we obtained here, instead we meant to use "predict" in a temporal sense (the change of association which then leads to a measurable change in behavior later on). We apologize for this too lenient usage of the word "predict" and have therefore removed the word "prediction" throughout the entire manuscript whenever it indeed referred to a correlation.

More minor points:

M1) *“Some of the tests are one sided, while others are two-sided. This seems justified, but these distinctions and justifications are somewhat buried in the methods section where many readers will miss them. I think it would be useful to mark the 1 and 2-sided tests where they appear in the results section.”*

Thanks for pointing this out. We agree that it is important to make it more evident to the reader which tests are one-tailed and why. We have updated the reported analyses and now consistently report one-tailed tests whenever we had a directed hypothesis. The directionality of the hypothesis is then more clearly introduced before the respective analysis and in the methods section.

M2) *“Is the following an error? That seems likely given the exact same values – although it may be that they are indeed the same. “we found decreased associative strength from CS0A ($Z = 2.27$, $P = 0.023$, Wilcoxon signed-rank test) and increased associative strength for CS+A ($Z = 2.27$, $P = 0.023$, Wilcoxon signedrank test) during POST” “*

Thanks for your careful reading. Both analyses indeed yielded the same statistical values (with effect sizes of different magnitude). We have re-run the analysis to double check. Please see the new paragraph below, now including effect sizes.

[...] we found decreased repetition suppression for CS^0_A ($Z = 2.26$, $P = 0.012$, $U3_1 = .67$, one-sample Wilcoxon signed-rank test, one-tailed) and increased repetition suppression for CS^+_A ($Z = 2.26$, $P = 0.012$, $U3_1 = .69$, one-sample Wilcoxon signed-rank test, one-tailed) during POST [...]

M3) *“What exactly is the difference between Fig 3B and S2A? They both seem to have the same contrast (they even specify eq. 3) but different values. Please clarify.”*

Our apologies, this was a typing mistake. We had incorrectly specified Equation 3 underlying the correlations depicted in Fig. 3B (now Figure 4B) and S2A (now S4A). However, Figure 3B (now Figure 4B) actually depicts the correlation between the contrast specified in Equation 2 (strengthening of CS^+_A relative to CS^+_B) and the overall choice probability for CS^+_A minus the overall choice probability for CS^+_B . Figure S2A in the previous submission (now Figure S4A) shows the correlation between the contrast specified in Equation 2 (strengthening of CS^+_A relative to CS^+_B) and the pairwise choice probability of CS^+_A vs CS^+_B , so only those binary choices directly contrasting CS^+_A and CS^+_B (as in revised Fig. S2H). We have changed the x-axis label in the revised Figure S4A to make this clearer. Sorry for the confusion.

M4) *“This sentence refers to the wrong equation, “We investigated post choice induced revaluation conjunction effects (Equation 9)”*

This should indeed have referred to equation 3. We have corrected this accordingly. Thank you for your careful reading and for spotting this.

Reviewer #2

„This is a well-designed study that investigates an important research question with implications for real-life decision-making problems that humans face every day. The design is intricate and sometimes a bit hard to follow, but it follows a rational logic tailored to the phenomenon of repetition suppression. I think the research question is broad enough to be interesting and relevant for the diverse readership of Nature Communications even outside the field of Cognitive Neuroscience. The analyses are planned with great care to the details and the rational approach provides a good thread to follow the logic of the design.“

We thank the reviewer for the very positive evaluation of our work, and particularly for the detailed, thoughtful comments and suggestions to report additional statistical parameters and to strengthen links to the existing literature.

1) *“First, some of the behavioral (but also the neuroimaging) effects appear to be not as strong given that they become significant with one-tailed testing. While there is nothing wrong with this in principle, I think the use of one-tailed test needs better preparation and justification in the description of the results and also in the Methods section, i.e. the directionality of the hypothesis needs to be stated more clearly and then should be followed through in all contrasts. Right now, the selection of one-tailed or two-tailed tests appears to be mainly driven by the size of the effect (if strong, then two-tailed test are reported, if weak, then the authors use one-tailed tests). This needs to be stratified. Another consideration that always comes up when using one-tailed tests is that of statistical power. It would be good, if the authors could provide a measure of effect size in the main text along with a measure of the (post-hoc) statistical power, so that the reader can get an impression of the strength of the effect was influenced by the power in the experiment for both the behavioral and neuroimaging data since one-tailed tests are also prevalent in the ROI analyses of the brain data.”*

Thank you for bringing this up. It is important that the justification for one-tailed tests (wherever applied) is evident to readers (please also see our response to reviewer #1, minor point 1). We have updated all of our analyses throughout the manuscript and now consistently report one-tailed tests whenever we had a directed hypothesis. We also introduce the directionality of the hypotheses more clearly before the respective analyses and in the methods section. Additionally, we now provide measures of effect sizes for all statistical tests performed and report power ($1-\beta$) for repeated-measures ANOVAs (specifically, interaction effects) and Wilcoxon signed-rank tests/t-tests, in case these had been the primary analyses. No power is reported for post-hoc tests.

2) *“Second, the fMRI experiment different from the preceding behavioral experiment in the introduction of the RS runs PRE and POST choice revaluation. In these runs, no outcomes are presented following the respective CS presentations. This open the possibility for extinction processes to kick in during these runs that could affect the changes induced by choice revaluation. Such possible confounds should be critically examined in the Discussion.”*

We thank the reviewer for this important concern. Extinction could have been a confounding factor in the current design. However, we believe that extinction effects are unlikely to explain the observed fMRI results. In our view, there are three arguments that speak against an explanation based on extinction effects. Firstly, during RS PRE and POST, the outcomes (US) were indeed presented following the CS (although these were "wrong" in 2/3 of the trials) and participants had to perform memory judgements based on their ability to distinguish “correct” associations from “incorrect” associations. Performance in these memory probe trials was generally high (overall: $M = .92$, $SD = .06$), without any indications of differences between PRE: $M = .92$, $SD = .074$ and POST: $M = .92$, $SD = .072$. Secondly, we would expect extinction effects to equally affect all of the acquired CS-US associations, not just the two critical ones ($CS^0_A - US^0$ and $CS^+_A - US^+$). However, in the hippocampus and orbitofrontal cortex there was no evidence for a PRE to POST change of the difference between the representation of the two control CS-US associations ($CS^-_A - US^-$ and $CS^-_B - US^-$). Finally, extinction would predict equidirectional changes of all CS-US associations from PRE to POST, which is not supported by the data. We are therefore confident that extinction cannot readily explain the observed opposing effects (strengthening of CS^+_A relative to CS^+_B and weakening of CS^0_A relative to CS^0_B). We think it is important that this is evident to readers and have therefore added the following paragraph (at the end of the newly named section “Choices modify univariate neural measures of stimulus-outcome associations”):

Importantly, the observed dissociation of choice-induced increase of RS effects for CS^+_A choice-induced decrease of RS effects for CS^0_A and the absence of PRE-POST differences of RS effects for the CS^-_A relative to CS^-_B cannot be readily explained by general extinction effects resulting from exposition to CS-US associations other than the initially learned associations. Extinction would have been expected to equally spread across all CS-US associations and would imply equidirectional changes of all CS-US associations from PRE to POST, which is incompatible with the observed results.

3) *“Third, this experiment connects to Tom Schonberg work on preference change through mere choices in a significant way, although this study is more multi-dimensional in its design*

and implications. Nevertheless, it would be good to connect to this literature and highlight the similarities and differences between Schonberg's approach and the present study."

We thank the reviewer for pointing us to the important work by Tom Schonberg and colleagues, cued approach training (CAT) effects in particular. Indeed, CAT effects are conceptually closely linked to choice-induced revaluation, especially since both effects are instances of non-reinforced preference change. However, we believe that the present study offers a memory-based mechanism underlying choice-induced preference change that goes beyond a trained action tendency or choice rule ("go" response), as observed in CAT. We have amended a section highlighting similarities and differences between CAT and the present work in the discussion (reproduced below in blue for convenience). Further to this, we also believe that your point is closely related to comment 4 by reviewer #3, and would therefore like to point you to our response to that comment (which also includes data from a new behavioral experiment, that is also presented in Figure 5).

Our results are in line with previous reports of choice-induced preference changes (Ariely & Norton, 2008; Brehm, 1956; Izuma et al., 2010; Sharot, Velasquez, & Dolan, 2010) and conceptually replicate studies showing valuation change of stimuli by cued approach training (CAT) (Aridan, Pelletier, Fellows, & Schonberg, 2019; Botvinik-Nezer, Bakkour, Salomon, Shohamy, & Schonberg, 2019; Salomon, Botvinik-Nezer, Oren, & Schonberg, 2019; Schonberg et al., 2014). Similar to the present approach, performance of a button press ("go" response) upon presentation of "go" stimuli during CAT reliably induces long-term (Salomon et al., 2019) non-reinforced changes of desirability and choice probabilities of "go" stimuli over "no-go" stimuli. CAT effects are independent of initial value of the stimuli and seem to depend on integrity (Aridan et al., 2019) and activation (Salomon et al., 2019; Schonberg et al., 2014) of ventromedial prefrontal cortex, and interactions between orbitofrontal cortex and ventral striatum (Salomon et al., 2019). Importantly, the results of Experiment 5 suggest that choice-induced revaluation effects, at least for previously chosen options, seem to go beyond a trained action tendency or choice rule ("go" response), as observed in CAT.

Minor points:

M1) *"In the description of the fMRI findings (line 171 and onwards), the contrast estimates are named "associative strength". This seems to be a bit suggestive cognitive process (i.e. you cannot directly measure associative strength through contrasting two BOLD activations). A*

straight description of the interaction would be more appropriate, which then be interpreted as associative strength."

Our apologies, we had indeed taken an interpretative leap here. We agree that it is better to use terminology that is less suggestive. Therefore, we have replaced "associative strength" by "repetition suppression" throughout the results section of the manuscript and in the figure labels. In the discussion, we then interpret the observed repetition suppression effects as associative strength.

M2) *"line 567: "participant was included"-> remove "was" "*

Thank you. Done.

M3) *"line 586: please spell out the meaning of COPE"*

Apologies, we had not noticed we were using FSL lingo. We now spell out the acronym "COPE" (contrast of parameter estimates) before its initial introduction.

Reviewer #3

“The paper is well written, coherently argued to the largest degree (see comments below though), and certainly an interesting read. Clear predictions are made, and tested with appropriate methods.”

We thank the reviewer both for the overall positive evaluation of our manuscript and for the detailed and thoughtful comments that prompted various additional analyses, a behavioral control experiment, and helped us clarify the hypothesized memory mechanism underlying choice-induced revaluation effects.

1) *“To analyse the choice biases behaviourally, the authors report that “For the decision probe phase, we computed an average overall CP per conditioned stimulus per subject including all binary decisions in which the respective CS was present” (p.20/21). I would like to see this behavioural data unpacked a little. In particular, I am assuming that the major effect will come from probe trials where subjects have to make a choice between two same-value CS, because otherwise they will choose the higher-valued CS on the vast majority of trials? At least it would be surprising if the strengthening/weakening was pronounced enough to override the conditioned value (e.g. when subjects select between CS with different associated values, e.g. CS+ and CS0). Relatedly, is the choice bias mainly produced by the 8 repetitions where participants directly choose (again) between the two CS that were the critical stimuli in the previous revaluation phase? If not, what types of trials does it mainly generalize to?”*

Thank you for making us elaborate on this. Our intention in presenting the overall CP had been to show the most robust/well-powered behavioral effects that could be achieved in the present design (i.e. an analysis incorporating all available choice trials). We completely agree with the reviewer that it would indeed have been surprising if the choice-induced revaluation effect had been able to "override" the outcome values to which the stimuli had initially been conditioned. As can be seen in the new Supplementary Figure S2, which we reproduce below for convenience (Figure 2), the reviewer correctly assumed that participants chose the higher-value CS in most of the trials and there were no indications of choice-induced "overriding" of initially conditioned values. While it is plausible to assume that the choice bias is primarily driven by the pairwise decisions of the respective critical CS against the same-value CS, such a revaluation act could still act across stimuli of different valence - e.g. participants would prefer CS⁺ over CS⁰ and CS⁻ (the lower value CS), but this difference would be more pronounced for the revalued CS⁺_A. Such effects would contribute to the overall CP, in addition to what can be obtained by the direct comparison of same-value CS. However, we agree with you that the pairwise choice effects in particular provide the most specific insights into the occurrence of

choice-induced preference changes (as they display the most relevant comparison of the critical CS against same-value CS). We therefore now report the statistical tests for these comparisons in the manuscript. It should be noted that these effects, unlike the overall CP, are based on a subset of trials and therefore yield lower statistical power. Accordingly, we could not reliably detect significant deviations from the indifference criterion of $CP = 0.50$ for all critical choice probabilities.

Furthermore, when comparing choice probabilities for the two (three in Experiment 3) CS that had been presented during the previous revaluation phase (“critical” CS, e.g. CS^+_A vs CS^0_A in Experiment 1) and the same decision between same-valued CS (e.g. CS^+_B vs CS^0_B in Experiment 1), we obtained the following results. Experiment 1: $Z = 3.32$, $P < .001$; Experiment 2: $Z = 3.02$, $P = .003$, Experiment 3: CS^-/CS^0 : $Z = 1.28$, $P = .200$, CS^+/CS^0 : $Z = .21$, $P = .837$; Experiment 4: $Z = 3.60$, $P < .001$. As expected, participants were consistent in their choice behavior across revaluation and decision probe phase.

Figure 2. Decision probe pairwise choice probabilities for CS that were presented during revaluation against every other CS. A) In Experiment 1, CS^0_A is only preferred over CS^-_A and CS^-_B (top), CS^+_A is the most preferred CS (bottom). B) In Experiment 2, CS^-_A is the least preferred CS (top), CS^0_A is chosen more frequently than CS^-_A , CS^-_B , and CS^0_B (bottom). C) In Experiment 4, CS^0_A is only preferred over CS^-_A and CS^-_B (top), CS^+_A is the most preferred CS (bottom). D) In Experiment 3, CS^0_A is chosen at the same frequency as CS^0_B (middle). E-H) Pairwise within-category choice probabilities displaying the pairwise comparison that are most indicative of choice-induced preference changes. E) Experiment 1, F) Experiment 2, G) Experiment 3, H) Experiment 4. These results suggest that initially conditioned value was not overridden by revaluation choices and that the choice bias was mostly driven by the pairwise decisions of the respective revaluation CS against the same-value CS. However, repetitions of choices between the two revaluation CS also contributed to the observed overall choice probability effects.

2) *“Regarding the fMRI data, the authors used a set of tightly controlled but quite complex contrasts, designed to isolate the specific effects of prior choice on associative strength, as measured by repetition suppression. It would be reassuring for the reader to see that within these complex contrasts, the effects in the hippocampus and OFC are driven by the expected components. That is, show (maybe in a supplementary figure) the degree of repetition suppression when the critical CS (e.g. CSA+ for strengthening) is paired with its original associate, compared to when it is shown with any of the other associates, separately for strengthening and weakening. If I understand Fig. 2 correctly, it depicts the difference between trials where the CSA was paired with its original outcome, and trials where the CSB was paired with its original outcome. Unpacking the pattern underlying these differences would help to determine whether there are any basic effects other than repetition suppression, e.g. exposure frequency, prediction error when CS are followed by the wrong outcome etc., that could influence the hippocampal signal. At current, I find it difficult to extract all these alternative possibilities from the data presented, and to see the direct evidence for mnemonic reactivation of the US.”*

We agree with the reviewer that the contrasts in the initial version of the manuscript were quite complex. In our view, this complexity was a necessary prerequisite for ruling out a number of confounding factors. In particular, the manner in which we had set up the contrasts made sure that stimulus-specific and outcome-related activations would cancel out. This allowed us to isolate the - likely more subtle - changes in associative strength that otherwise might be overshadowed by outcome- and stimulus-specific activations.

The average parameter estimates and Z-maps displayed in manuscript Figure 2 were based on the contrasts in equations 1 and 2, thus these effects represent choice-induced signal changes in the hippocampus and OFC for CS^0_A and CS^+_A , followed by the respective correct outcome (US^0 and US^+) – controlling for activation elicited by CS^0_A followed by both incorrect outcomes (US^- and US^+) and CS^+_A followed by both incorrect outcomes (US^- and US^+). Additionally, the term on the right-hand side controlled for the exact same patterns of activation elicited by the respective same-valued stimuli, CS^0_B and CS^+_B , respectively. We thus reasoned that, based on the sign of the resulting parameter estimate, the difference of repetition suppression would be interpretable as strengthening or weakening of the respective CS-US association relative to the same-value CS-US association. We apologize for not having elaborated enough on this complex contrast in the previous version of our manuscript. We now provide a more detailed explanation in the methods section and have adjusted the caption of Figure 2 accordingly.

Figure 3. Repetition suppression for $CS^0_A-US^0$, controlling for activation elicited by CS^0_A followed by both incorrect outcomes (US^- and US^+) (Equation 11) and $CS^+_A-US^+$ repetition suppression, controlling for activation elicited by CS^+_A followed by both incorrect outcomes (US^- and US^0) (Equation 12), in the left hippocampus (A) and right lateral orbitofrontal cortex (B).

Regarding your request to *"Unpack the pattern underlying these differences"*, we have run additional repetition suppression-based analyses that aimed at isolating the effects of weakening of CS^0_A (Equation 11) and strengthening of CS^+_A (Equation 12). The results (please see Figure 3 above) qualitatively capture the expected weakening of CS^0_A and strengthening of CS^+_A . However, we believe that caution is necessary when interpreting these results. This is because by breaking up the contrasts into "isolated" effects of CS^0_A and CS^+_A followed by the correct outcome, controlling for activation elicited by the CS followed by its respective incorrect US, we lose important specificity of the previously used contrasts. In other words, while still controlling for stimulus-specific effects, our potential changes in associative strength might now reflect outcome-related activation, which is now no longer controlled for. Perhaps less importantly, but still relevant, these contrasts are only based on half of the trials compared to the initially included contrasts, reducing statistical power. Thus, we believe that the initially included contrasts – whilst perhaps less intuitive – present more rigorously controlled evidence for the proposed mechanism of choice-induced changes to associative strength. We have therefore not included the figure above (Figure 3) in our revised manuscript but would of course be happy to do so if the reviewer thinks it might be useful.

CS⁰_A weakening contrast:

$$[2 \times CS^0_{A-US^0}] - [CS^0_{A-US^-} + CS^0_{A-US^+}] \quad (11)$$

CS⁺_A strengthening contrast:

$$[2 \times CS^+_{A-US^+}] - [CS^+_{A-US^-} + CS^+_{A-US^0}] \quad (12)$$

3) *“Related to the previous point, the authors make the strong claim that changes in associative strength underlie the observed effects. Given this aspect seems to be critical in terms of novelty, it would be reassuring to see conceptually similar results replicated with a complementary analysis, e.g. a multivariate analysis that directly decodes the (presumably mentally reinstated) outcome during each CS presentation. Of course, on correct trials of the fMRI-RS task, the outcome will be visually presented and thus strongly present in brain activity. there are plenty of repetitions in this task, however, where incorrect outcomes are shown, and where we would still expect participants to associatively retrieve the correct outcome, given they have to make correct-incorrect judgments during this phase. This could be shown e.g. using a representational similarity approach comparing the neural patterns elicited during CSA and CSB trials (minus the ones that were followed by the original outcome), or using pattern classifiers with the same underlying logic.”*

We would like to thank the reviewer for this thoughtful suggestion that inspired novel RSA-based analyses. Since we had not measured an independent localizer task for the US stimuli before learning CS-US associations, and since there was no possibility to obtain unbiased and independent multivariate patterns to be used in a leave-one-run-out cross-validation-based classification analysis, we chose a representational similarity approach.

As suggested by the reviewer, we investigated the PRE to POST changes in BOLD pattern similarity (as measured with Pearson’s product-moment correlation coefficient r) between CS⁰_A/CS⁰_B and CS⁺_A/CS⁺_B followed by incorrect outcomes (US⁻/US⁺ or US⁻/US⁰, respectively) in the left hippocampus and the right OFC. Across all CS pairs, the neural pattern similarity decreased from PRE to POST, an effect that was not present for the pair of control stimuli CS⁻_A/CS⁻_B (followed by US⁰/US⁺), which, if anything, tended to become more similar. These analyses led to the entirely new paragraph “Choice-induced decrease of multivariate neural pattern similarity” in the results section and Figure 4 below, which is included as new Figure 3 in the manuscript. We believe that these results significantly corroborate our findings based on repetition suppression effects and strengthen the conclusions that can be drawn from our data.

Figure 4. Multivariate Neural Pattern Similarity Analysis Results.

A) Multivariate neural pattern similarity analyses in the left hippocampus and B) right lateral OFC, showing that neural similarity (POST r – PRE r , Δ Pearson's r) between same value stimulus-outcome pairs ($CS^0_A-US^-/CS^0_B-US^-$ and $CS^0_A-US^+/CS^0_B-US^+$; $CS^+_A-US^-/CS^+_B-US^-$ and $CS^+_A-US^0/CS^+_B-US^0$) decreases from PRE to POST. Importantly, change of neural pattern similarity for the control stimulus pairs CS^-_A/CS^-_B is numerically positive, indicating higher similarity from PRE to POST.

We are therefore very grateful for the reviewer's suggestion. Please note however that this pattern similarity-based approach should be seen as complementary to our previous analyses. This is because, unlike the repetition suppression analysis initially reported, the neural pattern similarity analysis does not allow to interpret the directionality of changes in neural representations: any observed decrease in pattern similarity could result from either strengthening or weakening of associative memory traces. In both cases, measures of changes in neural similarity would be negative. Further to this, the fMRI PRE and POST runs were not initially designed for a pattern similarity analysis (or other multivariate methods), and therefore the analyses had to be based on a reduced subset of trials, generally under-powering the analyses ($1-\beta < .80$).

4) *"In some instances, I found it difficult to follow the exact reasoning regarding the hypothesized memory mechanism underlying the observed effects, and the paper could be improved if a more explicit and coherent explanation of this mechanism was offered. Based on the memory literature, I initially found it surprising, in the context of this study design, that the authors argue for a weakening of the unselected CS-US association, since the respective CS are still presented multiple times during the revaluation phase, presumably encouraging*

participants to repeatedly recall the associated values of both choice stimuli (e.g., CS+A and CS0A in Exp. 1 and 4) in order to make a value-based choice between them. In theory, repeated recall of both associations should lead to strengthening of both associations. Is it possible that participants simply learn a “do-not-choose-this-stimulus” tag along with the unselected CS during revaluation, rather than weakening the CS-US association of the unselected stimulus? I understand that the fMRI data do not support such a tagging explanation, but seeing additional (e.g. behavioural) evidence against it would be reassuring. Second, given the study design, it seems equally likely that repeatedly selecting CSA+ would weaken the CSB+ association, since the CSB+ is associatively linked to the same outcome and can thus be considered a competitor, maybe more so than the CSA0. Again, a more explicit argument regarding the specific memory mechanism, potentially supported by empirical evidence, would strengthen the conclusions that can be drawn from this study.”

If we correctly understand the reviewer, this comment actually relates to two different points, one related to the hypothesized mechanisms, the other to a potential alternative explanation of our data, based on learning a "tag".

4A) Hypothesized mechanism (please also see our response to reviewer #1, comment 3 above):

Our apologies for not being clear in the description of the hypothesized memory mechanism underlying the effect. We have updated the introduction with a more parsimonious explanation for both choice-induced strengthening and weakening of the associative traces. In naturalistic decision making scenarios, choices often have to be made without direct experience of feedback or rewards. Instead, decision makers have to rely on relational knowledge of stimuli, actions and outcomes. The absence of direct external feedback and the resulting inability to adjust synaptic weights based on error-driven learning mechanisms suggests the use of unsupervised learning to optimize behavior in those situations. Likely candidate mechanisms for such unsupervised behavioral adaptation are memory retrieval dynamics. It is well established that retrieval of an item from memory, e.g. a conditioned stimulus (CS) triggering retrieval of an associated outcome, leads to improved remembering. However, memory for competing items, e.g. a CS associated with the same outcome, is impaired simultaneously (Anderson et al., 1994; Hulbert & Norman, 2015; Wimber et al., 2015). Such retrieval-induced forgetting (Anderson et al., 1994) would predict choice biases towards previously chosen CS based on retrieval-related strengthening of CS-US association. However, retrieval-induced forgetting would predict the same effect for a previously presented, but unchosen CS, since both chosen and unchosen CS activate neural populations representing the respective associated outcome (Barron et al., 2013; Boorman et al., 2016; Howard et al., 2016; Klein-

Flugge et al., 2013; Onat & Büchel, 2015; Tonegawa et al., 2018). A more recent theoretical framework (Nonmonotonic Plasticity Hypothesis, as reviewed in Ritvo, Turk-Browne, & Norman, 2019) suggests U-shaped spreading activation during associative memory retrieval: Inactive memories remain unaltered, whereas moderately activated associative memories are weakened, and higher activation leads to strengthening of memories (Ritvo et al., 2019). Translating this idea to memory-based decisions between two CS, we assumed that both CS would moderately activate neural populations representing the associated outcome (as the outcome is never presented). However, consistent with the finding that chosen options receive higher attentional weighting than unchosen options (as reflected in higher learning rates for chosen options (Klein et al., 2017; Palminteri et al., 2015)), we further assumed that choices of a CS would induce additional activation of the associated outcome, whereas this would not be the case for unchosen CS, retaining an intermediate activation state of the associated outcome. Thus, we hypothesized that choosing a CS would strengthen the related stimulus-outcome association. Contrarily, not choosing a CS would weaken the respective stimulus-outcome association. We expected that these choice-induced alterations of the associative memory structure would result in subsequent preference changes. In other words, we assumed that choices themselves can act as self-generated “teaching signals”, dynamically altering stimulus-outcome associations stored in memory by shifting associative memories along a nonmonotonic plasticity function (Ritvo et al., 2019).

The RS contrasts we used explicitly allow for the interpretation of both strengthening of $CS^+_A-US^+$ or retrieval-induced forgetting of the competing association $CS^+_B-US^+$. Based on the initially used RS contrasts, both mechanisms make the same prediction and cannot be dissociated. However, we once again want to thank the reviewer for the helpful suggestion of showing simple contrasts separately for strengthening and weakening and conclude that there is at least qualitative evidence for choice-induced strengthening of $CS^+_A-US^+$ and weakening of $CS^0_A-US^0$ (see Figure 3).

4B) potential alternative explanations

This is a very important point. The reviewer essentially suggests that our behavioral pattern may as well emerge from participants learning a simple choice rule by “tagging” the chosen CS as a “Go stimulus” and the unchosen CS as a “No-Go stimulus”. As the reviewer already pointed out, the pattern of fMRI results makes this unlikely. Nevertheless, we agree with the reviewer that it would be much more convincing to have stronger behavioral evidence. Your comment made us think hard about how we could address this in our present data, and we reached to the conclusion that an additional behavioral experiment would be required to arbitrate between these two alternative explanations. This experiment (and the results of it) is

described in detail on p. 8 and p. 25-27 and p. 30-31 in the revised version of the manuscript, but let us briefly lay it out here:

Experiment 5 followed the same experimental logic as the previous experiments. To orthogonalize contributions of “go” and “no-go” tagging and associative strength between CS and US to choice probabilities, we let participants learn associations between 4 neutrally rated CS and an intermediate value outcome and a high value outcome (with two CS linked to an intermediate, and two CS linked to a high value outcome). Importantly, one of the two CS in each pair was followed by the outcome with a probability of 20% while the other CS was followed by the US in 80% of the trials. In the revaluation phase, participants chose between an intermediate and a high value CS, always being presented with choices between CS^+_{80} versus CS^0_{80} and CS^+_{20} versus CS^0_{20} . Again, no outcomes were presented at this stage. In the subsequent decision probe phase, participants made choices between the two same-value pairs of stimuli that differed in terms of how strongly they were associated with their respective outcomes.

According to our hypothesis (“associative account”) probe phase decisions are guided by learned associations and strengthening/weakening of this association during revaluation. Therefore, we expected (after revaluation) a significantly increased choice probability for both highly associated stimuli: CS^0_{80} should be preferred over CS^0_{20} , and CS^+_{80} should be preferred over CS^+_{20} . On the contrary, if choice behavior was exclusively driven by learned “go” and “no-go” tags, and participants had learned “go” tags for both chosen CS^+_{80} and CS^+_{20} and “no-go” tags for both unchosen CS^0_{80} and CS^0_{20} (instead of changing the underlying associative structure), both same-value pairwise choice probabilities should be at chance level (CP = 0.50). The results of Experiment 5 are summarized in Figure 5. Importantly, there was no significant difference between revaluation choice probabilities of CS^+_{80} versus CS^0_{80} and CS^+_{20} versus CS^0_{20} ($Z = 1.19$, $P = .234$, $U3 = .53$, Wilcoxon signed-rank test, two-tailed), ruling out of unequal assignment of “go” and “no-go” tags across CS pairs. We observed that participants favored the previously chosen (“go” tag) and strongly associated CS^+_{80} over the previously chosen and weakly associated CS^+_{20} ($Z = 3.55$, $P < .001$, $U3_1 = .75$, $1-\beta > .99$, one-sample Wilcoxon signed-rank test, one-tailed). This pattern of results favors an explanation based on associative strengthening of the memory trace between CS^+ and US^+ , rather than merely on expressing a “go tag”. Descriptively, participants also tended to favor the previously unchosen (“no-go” tag) and strongly associated CS^0_{80} compared to the previously unchosen and weakly associated CS^0_{20} ($Z = 0.61$, $P = .271$, $U3_1 = .61$, $1-\beta = .23$, one-sample Wilcoxon signed-rank test, one-tailed, Fig. 5 below). Again, this pattern is not consistent with a “tagging-based” explanation.

Figure 5. Behavioral control experiment (Experiment 5), orthogonalizing contributions of “Go” and “No-Go” tagging and associative strength between CS and US to choice probabilities. Previously chosen (“Go” tag) and strongly associated CS⁺_{.80} is preferred over previously chosen and weakly associated CS⁺_{.20} (blue scatter), while there is only descriptive evidence for preference of previously unchosen (“No-Go” tag) and strongly associated CS⁰_{.80} over previously unchosen and weakly associated CS⁰_{.20}.

However, the absence of a clear preference for CS⁰_{.80} compared to CS⁰_{.20} does not provide definite evidence against the alternative explanation that participants learned “no-go” tag for the unchosen stimulus. This asymmetry in expression of response tendencies could result from differential mechanisms driving acquisition of “go” and “no-go” choice rules, akin to well-described Pavlovian biases (Guitart-Masip et al., 2012). Possibly, in high-value choice trials, the majority of participants used the learned and updated CS-US associative strength instead of “go” response tendencies to guide their decisions, while this was only descriptively the case for intermediate-value choices (*Median* choice probability = .60, *range* = 0 – 1).

Presumably, reverting the initially learned “go” response tendency for CS⁺_{.20} could have less of an impeding effect on re-acquisition of a “no-go” response during decision probe phase, than re-acquisition of a “go” response for CS⁰_{.80} initially learned with a “no-go” choice rule, consistent with proposals of asymmetric instrumental learning of action and inaction (Swart et al., 2017). Altogether, while we believe that the present set of findings makes it unlikely that our data can primarily be explained by such a heuristic account, we also concede that it may contribute to the “no-go”-bias observed for the unchosen stimuli in our experiments. Again, we are grateful to the reviewer for raising this important point. We think that the results we obtained in response to this further strengthen the conclusions that can be drawn.

5) *“In the fMRI-RS task, participants were asked to make memory judgments on whether the CS-US combination matched the originally learned pairing. It would be interesting to know if the behavioural data during this phase provide any evidence for strengthening/weakening of the critical, revalued associations, either in terms of percent correct (which might be at ceiling, with the reported mean of 92%), or in terms of reaction times.”*

Indeed, there was no evidence for changes in terms of percentage correct responses for the critical CS from PRE to POST in the memory probe trials. Performance (percentage correct answers) was generally very high and presumably at ceiling across all CS, PRE and POST (all $M_s > .91$). We did not observe significant within value category differences between PRE to POST differences (all $t_s < .67$, $P_s > .563$). However, for CS^0_A we observed a descriptive increase of RT for correct responses from PRE to POST when excluding an outlier with a change of RT at > 3 SD below the mean ($M = .02$, $SD = .16$; including outlier: $M = .003$, $SD = .20$). All other CS showed a descriptive decrease of RT from PRE to POST, which would have been expected, as it presumably reflects participants' getting better at performing the task (e.g. getting used to the timing and presentation format). When excluding the outlier, we found a marginal effect for the RT change between CS^0_A and CS^0_B ($t_{39} = 1.82$, $P = .076$), which did not hold when including the outlier ($t_{41} = 1.29$, $P = .205$). For all other stimuli, there were no within-category differences of PRE to POST change of RTs (all $t_s < .54$, $P_s > .601$).

To us, this pattern of results does not provide clear evidence either for or against weakening/strengthening of the CS-US association. However, firstly, there is only a limited number of trials per stimulus that enters this analysis, which renders them potentially underpowered. Secondly, the fact that making a correct/incorrect judgement on a given association (as opposed to making a value-based decision between options) may also play a role here. We have therefore not included these analyses in the manuscript for now, but would of course be happy to do so, if the reviewer thinks they might be useful.

6) *“The behavioural data from the fMRI experiment do not replicate the behavioural effect found in Exp. 1 and 2, in particular the weakening of the unselected CS. The authors suggest that this null effect might be due to reconsolidation via re-exposure during the fMRI-RS task. It is difficult to perceive how reconsolidation could play a role at these short, within-session time scales, given that reconsolidation in the strict sense can only occur for memories that were consolidated (e.g. by 48h delay, sleep etc.) in the first place. The authors should at minimum elaborate on this explanation. A more parsimonious, alternative view is that, consistent with the memory literature (e.g. Hulbert & Norman, CerebCortex; Storm, Bjork & Bjork, 2008), is that retrieval-induced weakening can be eliminated or even reversed into strengthening when interleaved with, or followed by, re-exposure to the weakened representations.”*

Indeed, it is highly unlikely that the observed absence of reduced choice probability for CS^0_A in the fMRI experiment can be attributed to reconsolidation. We want to apologize for this inaccurate interpretation. We have removed any reconsolidation-based interpretation from the manuscript. We want to thank the reviewer for offering the far more parsimonious explanation of these effects by re-exposure to the initially learned association, as seen in reversal of retrieval-induced forgetting by introduction of a re-study phase. We have modified the discussion on the observed absence of reduced choice probability for CS^0_A accordingly:

To assess PRE to POST changes of associative strength participants in the present study had to be re-exposed with the initially learned CS-US associations and were explicitly instructed to judge whether the presented CS-US associations were correct. It is well established that restudying of memorized material reverses retrieval-induced forgetting effects (Hulbert & Norman, 2015; Storm, Bjork, & Bjork, 2008). Thus, the observed null effect for CS^0_A might be due to re-exposure and restudy of the original CS-US association. This process might have allowed the weakened association between CS^0_A - US^0 to regain its original associative strength.

Minor comment:

M1) *“Figure 2 would benefit from a visualization of which comparisons (interaction, paired tests) were significant in the bar graphs shown at the bottom.”*

We have added a visualization of significant comparisons to the revised Figure 2, as requested.

References

- Anderson, M. C., Bjork, R. A., & Bjork, E. L. (1994). Remembering Can Cause Forgetting: Retrieval Dynamics in Long-Term Memory. *Journal of Experimental Psychology: Learning, Memory, and Cognition*, *20*(5), 1063–1087. <https://doi.org/10.1037/0278-7393.20.5.1063>
- Aridan, N., Pelletier, G., Fellows, L. K., & Schonberg, T. (2019). Is ventromedial prefrontal cortex critical for behavior change without external reinforcement? *Neuropsychologia*, *124*(December 2018), 208–215. <https://doi.org/10.1016/j.neuropsychologia.2018.12.008>
- Ariely, D., & Norton, M. I. (2008). How actions create - not just reveal - preferences. *Trends in Cognitive Sciences*, *12*(1), 13–16. <https://doi.org/10.1016/j.tics.2007.10.008>
- Barron, H. C., Dolan, R. J., & Behrens, T. E. J. (2013). Online evaluation of novel choices by simultaneous representation of multiple memories. *Nature Neuroscience*, *16*(10), 1492–1498. <https://doi.org/10.1038/nn.3515>
- Boorman, E. D., Rajendran, V. G., O'Reilly, J. X., & Behrens, T. E. (2016). Two Anatomically and Computationally Distinct Learning Signals Predict Changes to Stimulus-Outcome Associations in Hippocampus. *Neuron*, *89*(6), 1343–1354. <https://doi.org/10.1016/j.neuron.2016.02.014>
- Botvinik-Nezer, R., Bakkour, A., Salomon, T., Shohamy, D., & Schonberg, T. (2019). Memory for Individual Items is Related to Non-Reinforced Preference Change. *BioRxiv*, 621292. <https://doi.org/10.1101/621292>
- Brehm, J. W. (1956). Postdecision changes in the desirability of alternatives. *Journal of Abnormal and Social Psychology*, *52*(3), 384–389. <https://doi.org/10.1037/h0041006>
- Guitart-Masip, M., Huys, Q. J. M., Fuentemilla, L., Dayan, P., Duzel, E., & Dolan, R. J. (2012). Go and no-go learning in reward and punishment: Interactions between affect and effect. *NeuroImage*, *62*(1), 154–166. <https://doi.org/10.1016/j.neuroimage.2012.04.024>
- Howard, J. D., Kahnt, T., & Gottfried, J. A. (2016). Converging prefrontal pathways support associative and perceptual features of conditioned stimuli. *Nature Communications*, *7*(May), 1–11. <https://doi.org/10.1038/ncomms11546>
- Hulbert, J. C., & Norman, K. A. (2015). Neural differentiation tracks improved recall of competing memories following interleaved study and retrieval practice. *Cerebral Cortex*, *25*(10), 3994–4008. <https://doi.org/10.1093/cercor/bhu284>
- Izuma, K., Matsumoto, M., Murayama, K., Samejima, K., Sadato, N., & Matsumoto, K. (2010). Neural correlates of cognitive dissonance and choice-induced preference change. *Proceedings of the National Academy of Sciences*, *107*(51), 22014–22019. <https://doi.org/10.1073/pnas.1011879108>
- Klein-Flugge, M. C., Barron, H. C., Brodersen, K. H., Dolan, R. J., & Behrens, T. E. J. (2013). Segregated Encoding of Reward-Identity and Stimulus-Reward Associations in Human Orbitofrontal Cortex. *Journal of Neuroscience*, *33*(7), 3202–3211. <https://doi.org/10.1523/JNEUROSCI.2532-12.2013>
- Klein, T. A., Ullsperger, M., & Jocham, G. (2017). Learning relative values in the striatum induces violations of normative decision making. *Nature Communications*, *8*, 1–12. <https://doi.org/10.1038/ncomms16033>
- Onat, S., & Büchel, C. (2015). The neuronal basis of fear generalization in humans. *Nature Neuroscience*, *18*(12), 1811–1818. <https://doi.org/10.1038/nn.4166>
- Palminteri, S., Khamassi, M., Joffily, M., & Coricelli, G. (2015). Contextual modulation of value signals in reward and punishment learning. *Nature Communications*, *6*. <https://doi.org/10.1038/ncomms9096>
- Ritvo, V. J. H., Turk-Browne, N. B., & Norman, K. A. (2019). Nonmonotonic Plasticity: How Memory Retrieval Drives Learning. *Trends in Cognitive Sciences*, *23*(9), 726–742. <https://doi.org/10.1016/j.tics.2019.06.007>
- Salomon, T., Botvinik-Nezer, R., Oren, S., & Schonberg, T. (2019). Enhanced striatal and prefrontal activity is associated with individual differences in nonreinforced preference change for faces. *Human Brain Mapping*, (September), 1–18. <https://doi.org/10.1002/hbm.24859>

- Schonberg, T., Bakkour, A., Hover, A. M., Mumford, J. A., Nagar, L., Perez, J., & Poldrack, R. A. (2014). Changing value through cued approach: An automatic mechanism of behavior change. *Nature Neuroscience*, *17*(4), 625–630. <https://doi.org/10.1038/nn.3673>
- Sharot, T., Velasquez, C. M., & Dolan, R. J. (2010). Do decisions shape preference? Evidence from blind choice. *Psychological Science*, *21*(9), 1231–1235. <https://doi.org/10.1177/0956797610379235>
- Storm, B. C., Bjork, E. L., & Bjork, R. A. (2008). Accelerated Relearning After Retrieval-Induced Forgetting: The Benefit of Being Forgotten. *Journal of Experimental Psychology: Learning Memory and Cognition*, *34*(1), 230–236. <https://doi.org/10.1037/0278-7393.34.1.230>
- Swart, J. C., Froböse, M. I., Cook, J. L., Geurts, D. E. M., Frank, M. J., Cools, R., & den Ouden, H. E. M. (2017). Catecholaminergic challenge uncovers distinct Pavlovian and instrumental mechanisms of motivated (in)action. *ELife*, *6*, 1–36. <https://doi.org/10.7554/eLife.22169>
- Tonegawa, S., Morrissey, M. D., & Kitamura, T. (2018). The role of engram cells in the systems consolidation of memory. *Nature Reviews Neuroscience*, *19*(8), 485–498. <https://doi.org/10.1038/s41583-018-0031-2>
- Wimber, M., Alink, A., Charest, I., Kriegeskorte, N., & Anderson, M. C. (2015). Retrieval induces adaptive forgetting of competing memories via cortical pattern suppression. *Nature Neuroscience*, *18*(4), 582–589. <https://doi.org/10.1038/nn.3973>

Reviewers' Comments:

Reviewer #1:

Remarks to the Author:

The authors have addressed all of my concerns.

Reviewer #2:

Remarks to the Author:

The author have provided a comprehensive revision with additional new data that addresses all my questions. I support the publication now. Congrats.

Reviewer #3:

Remarks to the Author:

The authors revised the manuscript extensively, and were very responsive to the comments raised by the 3 reviewers. The revision includes a number of additional analyses and one new behavioural experiment, and the manuscript has improved substantially as a result. I do have one major concern with respect to the multivariate analysis that was added, see comment 3 below, but otherwise feel that my concerns were addressed sufficiently. The following comments are sorted according to the original points raised.

(1) The new supplementary figure sufficiently addresses previous concern #1.

(2) Regarding previous concern #2, It is reassuring to see that the pre-to-post changes in hippocampal and OFC activity go in the expected direction. I understand the authors' concern about inclusion of this data in the main manuscript, given these contrasts are less tightly controlled. Having said that, I do see a benefit in including these values underlying the observed activity changes, and the authors may want to consider including the bar graphs either as an additional panel in their main figure, or as a supplementary figure.

(3) The authors added a new multivoxel pattern similarity analysis to complement the main repetition suppression results, in response to the concern that repetition suppression is quite an indirect measure of memory reactivation. The results of this new analysis are somewhat inconclusive. The authors compare changes in pre-to-post neural similarity (in HC and OFC) separately for CS-, CS0 and CS+ trials. They find that while the CS- pairs, acting as control items in the fMRI study, show a numerical increase in similarity, the CS0 and CS+ pairs show a decrease. The decrease for the CS+ is surprising in particular, and in my opinion does not support the associative strengthening claim. If the stimulus-outcome association for CS+ is strengthened during reevaluation, and this strengthening is reflected in increased reactivation of the associate in the POST test, wouldn't one expect that all CS+ pairs (or at least all CS+A pairs) are neurally more similar to each other in the POST (compared to the PRE) test? I find the new results difficult to reconcile with a strengthening view, and am unsure why the authors argue that both weakening and strengthening would lead to a decrease in similarity. The manuscript should at minimum offer a convincing explanation for the pre-to-post similarity decrease for all reevaluation items.

(4) Regarding the theoretical interpretation of the effect, the authors took my (and another reviewer's) comments very seriously, and have changed the theoretical framing of the hypothesized effects, as well as adding an additional behavioural experiment to rule out an alternative explanation.

While the additional experiment does not entirely rule out this concern, the authors appropriately discuss this alternative explanation in the light of their data, and it thus seems sufficiently addressed in the manuscript.

(5) I agree with the authors that it is not necessary to report these additional results in the manuscript.

(6) The authors changed their interpretation in line with my previous comment, and this concern is not appropriately addressed.

We would like to thank the reviewers for the time and effort they put into the assessment of our revised manuscript and the overall very positive evaluation of our work. Nevertheless, two remarks (one minor, one major) by Reviewer #3 still required further clarification. We reproduce these comments below in italics before our response to each point. Changes made to the manuscript are marked in blue, both here and in the manuscript.

2.) *“Regarding previous concern #2, It is reassuring to see that the pre-to-post changes in hippocampal and OFC activity go in the expected direction. I understand the authors’ concern about inclusion of this data in the main manuscript, given these contrasts are less tightly controlled. Having said that, I do see a benefit in including these values underlying the observed activity changes, and the authors may want to consider including the bar graphs either as an additional panel in their main figure, or as a supplementary figure.”*

Thank you again for suggesting these additional analyses which provide further insights into the effects presented. We now present these results in the Supplementary Materials as new Supplementary Figure S3.

3.) *“The authors added a new multivoxel pattern similarity analysis to complement the main repetition suppression results, in response to the concern that repetition suppression is quite an indirect measure of memory reactivation. The results of this new analysis are somewhat inconclusive. The authors compare changes in pre-to-post neural similarity (in HC and OFC) separately for CS-, CS0 and CS+ trials. They find that while the CS- pairs, acting as control items in the fMRI study, show a numerical increase in similarity, the CS0 and CS+ pairs show a decrease. The decrease for the CS+ is surprising in particular, and in my opinion does not support the associative strengthening claim. If the stimulus-outcome association for CS+ is strengthened during revaluation, and this strengthening is reflected in increased reactivation of the associate in the POST test, wouldn’t one expect that all CS+ pairs (or at least all CS+A pairs) are neurally more similar to each other in the POST (compared to the PRE) test? I find the new results difficult to reconcile with a strengthening view, and am unsure why the authors argue that both weakening and strengthening would lead to a decrease in similarity. The manuscript should at minimum offer a convincing explanation for the pre-to-post similarity decrease for all revaluation items.”*

Our apologies for not having been clear enough about the reasoning behind our assumption of decreased PRE to POST similarity for both weakened and strengthened CS with their respective partner CS. We believe that there is indeed no plausible scenario under which one could obtain an increase in pattern similarity in our analyses:

Firstly, if during PRE, the CS⁺ *completely* reinstates the entire neural pattern caused by the US⁺, there is no room for enhanced reinstatement by choice-induced associative strengthening. In this case, there would be no net change in similarity. In our view, this scenario appears unlikely anyhow. If however, on the contrary, there is only partial reinstatement of the US⁺ pattern during PRE, and enhanced strength of the association leads to increased recruitment of elements of the original pattern by CS⁺_A (but not by non-presented CS⁺_B) during POST, then one would indeed expect *decreased* similarity. This latter scenario is consistent with the memory literature, specifically with Competitive Trace Theory (CTT, Yassa & Reagh, 2013) and with the Nonmonotonic Plasticity Hypothesis (Ritvo, Turk-Browne, & Norman, 2019). From this literature, it could be expected that during retrieval of US⁺ by presentation and decision for CS⁺_A, the memory trace between CS⁺_A and US⁺ would be strengthened in the revaluation phase. Again, this scenario would lead to a decrease of similarity between CS⁺_A and CS⁺_B, which is what we observed.

Secondly, another possibility is that there is no change in the association between CS⁺_A-US⁺, but instead in the association between CS⁺_B-US⁺. CS⁺_B had not been presented during revaluation. It is therefore unlikely that the memory trace between CS⁺_B and US⁺ would be actively rehearsed. Hence, this trace would be subject to passive decay, presumably resulting in a slightly weakened association from PRE to POST. Again, this would result in a decrease in similarity.

Additionally, consistent with the literature on retrieval-induced forgetting (Anderson, Bjork, & Bjork, 1994; Hulbert & Norman, 2015; Wimber, Alink, Charest, Kriegeskorte, & Anderson, 2015), it is plausible to assume that actively retrieving the memory trace between the target stimulus CS⁺_A and US⁺ will weaken the memory trace between the competitor stimulus CS⁺_B and US⁺. This would lead to a differentiation of the memory engrams encoding CS⁺_A and US⁺, and CS⁺_B and US⁺ - and hence again *diminished* similarity.

Thus, we acknowledge that the predicted (and observed) decrease in similarity could have resulted from either strengthening of the memory trace between CS⁺_A and US⁺ or from weakening of the memory trace between CS⁺_B and US⁺ – or a combination of both effects.

Indeed, the only scenario we could think of that could have led to *increased* similarity in our analyses would be that both of the two associations CS⁺_A-US⁺ and CS⁺_B-US⁺ are strengthened from PRE to POST. In this case, an enhanced reinstatement of the common component US⁺ would lead to increased similarity. However, we would argue that such a passive strengthening of the CS⁺_B-US⁺ association is not plausible – and would in fact speak against our results.

We have added one paragraph in the Methods section of the manuscript that is also referenced in the Results, elaborating on our assumptions on the RSA effects (and the directionality thereof), which we reproduce below:

The assumption of decreased neural pattern similarity for pairs of CS^+_A and CS^+_B as a result of choice-induced strengthening of CS^+_A is based on two grounds. Firstly, we assumed that both CS^+_A and CS^+_B would equivalently, and partially reinstate the pattern representing US^+ during PRE, but repeated retrieval of US^+ and choice of CS^+_A should selectively strengthen the association between CS^+_A and US^+ during POST. Since CS^+_B is not presented during reevaluation and thus not actively rehearsed, the memory trace between CS^+_B and US^+ should be subject to passive decay, presumably resulting in a slightly weakened association. Secondly, consistent with the literature on retrieval-induced forgetting (Anderson, Bjork, & Bjork, 1994; Hulbert & Norman, 2015; Wimber, Alink, Charest, Kriegeskorte, & Anderson, 2015) it is plausible to assume that actively retrieving the memory trace between the target stimulus CS^+_A and US^+ would additionally weaken the memory trace between the competitor stimulus CS^+_B and US^+ . Both of the above mechanisms would lead to a differentiation of the memory engrams encoding CS^+_A and US^+ , and CS^+_B and US^+ and should be reflected in diminished PRE to POST similarity.

Regarding the second part of the reviewer's comment (*wouldn't one expect that all CS+ pairs (or at least all CS+A pairs) are neurally more similar to each other in the POST (compared to the PRE) test?*), we further computed *within-CS* (CS^+_A) changes in similarity. We agree with the reviewer, that in stark contrast with the *between-CS* similarities discussed above, these *within-CS* similarities should yield increased similarity. We found that the neural pattern similarity between CS^+_A followed by US^- and CS^+_A followed by US^0 did not change significantly from PRE to POST (PRE to POST similarity change, Hippocampus: $M = -.006$, $SD = .094$, $t_{41} = 0.39$, $P = .701$; OFC: $M = -.020$, $SD = .14$, $t_{41} = 0.91$, $P = .368$). However, we think two issues should be considered when interpreting this lack of effect. Firstly, it might be related to low statistical power, as only a limited number of trials is included in these comparisons (correlation between 20 trials of each CS^+_A pair during PRE and POST, respectively). Secondly, and perhaps more importantly, we would assume that in these correlations, the strongest determinant is the US that is actually presented (e.g. US^0 vs US^- in the correlation between $CS^+_A-US^0$ and $CS^+_A-US^-$). Thus, the much more subtle contribution of stronger reinstatement of the common component US^+ from PRE to POST would be masked by the vastly different activity patterns caused by the different US.

For these reasons, in the manuscript, we prefer to not present these *within-CS* analyses, but instead opted for the *between-CS* approach currently presented in the manuscript. We would argue that computing pattern similarity (change) *between* both pairs of $CS^+_A-US^-$ and $CS^+_B-US^-$, and $CS^+_A-US^0$ and $CS^+_B-US^0$, allows us to control for the confounding effects of US^- or US^0 presentations (which are constant between conditions),

thus rendering them unlikely to contribute to the observed change in pattern similarity. We therefore believe that assessing pattern similarity *between* CS^+_A and CS^+_B (as opposed to *within*-CS similarity $CS^+_A-US^-$ and $CS^+_A-US^0$) affords more control over confounding factors contributing to (dis)similarity.

While the multivariate results on their own might not be compelling in their own right, we would also like to kindly note that the experiment was not designed for this analysis approach. However, together with our repetition suppression findings, we still think the multivariate results provide converging evidence to support our claims and hope the reviewer agrees with this.

References

- Anderson, M. C., Bjork, R. A., & Bjork, E. L. (1994). Remembering Can Cause Forgetting: Retrieval Dynamics in Long-Term Memory. *Journal of Experimental Psychology: Learning, Memory, and Cognition*, 20(5), 1063–1087. <https://doi.org/10.1037/0278-7393.20.5.1063>
- Hulbert, J. C., & Norman, K. A. (2015). Neural differentiation tracks improved recall of competing memories following interleaved study and retrieval practice. *Cerebral Cortex*, 25(10), 3994–4008. <https://doi.org/10.1093/cercor/bhu284>
- Ritvo, V. J. H., Turk-Browne, N. B., & Norman, K. A. (2019). Nonmonotonic Plasticity: How Memory Retrieval Drives Learning. *Trends in Cognitive Sciences*, 23(9), 726–742. <https://doi.org/10.1016/j.tics.2019.06.007>
- Wimber, M., Alink, A., Charest, I., Kriegeskorte, N., & Anderson, M. C. (2015). Retrieval induces adaptive forgetting of competing memories via cortical pattern suppression. *Nature Neuroscience*, 18(4), 582–589. <https://doi.org/10.1038/nn.3973>
- Yassa, M. A., & Reagh, Z. M. (2013). Competitive trace theory: A role for the hippocampus in contextual interference during retrieval. *Frontiers in Behavioral Neuroscience*, 7(JUL), 1–13. <https://doi.org/10.3389/fnbeh.2013.00107>

Reviewers' Comments:

Reviewer #3:

Remarks to the Author:

The authors have addressed all remaining concerns sufficiently in their revision, and I fully understand and agree with the explanations given the response letter for aspects (about the similarity-based analysis) that were not included in the final manuscript. I thus have no remaining concerns and congratulate the authors on an important piece of work.